# Untrackable distal ejecta on planetary surfaces

Rui Xu [1,3], Zhiyong Xiao [1,2,3] ✉, Fanglu Luo [1], Yichen Wang [1] & Jun Cui[1,2]

Impact ejecta are important references to establish regional and global stratigraphy of planetary bodies. Canonical views advocate radial distributions of distal ejecta with respect to the source crater, and their trajectories are significantly deflected on fast-rotating bodies. The Hokusai crater on Mercury formed a peculiar ray that features a hyperbola shape, and the sharp swerve of orientation was interpreted as a sign of a faster planetary rotation in the near past. Here, we show that this ray was not caused by a hypothesized larger Coriolis force, but due to abruptly-steepened ejection angles. Heterogeneous shock impedances of pre-impact impactor and/or target, such as topographic undulations, affect local propagation paths of shock and rarefaction waves, causing sudden changes of ejection angles. Distal ejecta with non-radial distributions are an inherent product of planetary impacts, and their unobvious provenances could mislead stratigraphic interpretations and hamper age estimations based on spatial densities of impact craters.

High-speed collisions of celestial materials are a ubiquitous geological process during the formation and evolution of planetary bodies. Impact cratering forms various forms of ejecta that can be deposited over far-flung areas, and those landed at more than about five radii of the source crater are termed as distal ejecta[1]. Distal ejecta are basic references used to apply the law of superposition on exterrestrial bodies, so regional and global stratigraphic systems can be established[2]. It is a conventional knowledge that ejection angles during impact cratering are usually 45° ± 15° (refs. [3–5]), and distal ejecta generally follow radial distributions[1]. Recent impact modeling revealed that topographic roughness of pre-impact target[6,7], inelastic collisions of impact ejecta[8–10], and oblique impact angles[11] could affect the overall radial distribution of impact rays. Meanwhile, it is well understood that the Coriolis force, especially on fast-rotating planetary bodies such as the Earth and Mars, possesses significant deflections on trajectories of distal ejecta, causing non-radial distributions[12–14].

On the airless planet Mercury, a hyperbola-shaped impact ray is prominent at the western hemisphere, which exhibits a sharp swerve in the orientation (Fig. 1). This ray was initially discovered in low-resolution images (~1 km/pixel) that were obtained by the Mariner 10 spacecraft[15], which covered less than half of the planet (Supplementary Fig. 1). Preliminary ballistic trajectory modeling for impact ejecta suggested that a recent impact at low latitudes of the eastern hemisphere formed this curved ray, when Mercury had a rotational period of about 6 Earth days[12] instead of the current 58.64 Earth days. This possibility is intriguing, because it remains an open question about when Mercury reached the 3:2 resonances that set up the current lithospheric thermal structure[16].

The eastern hemisphere of Mercury was not imaged by the Mariner 10 spacecraft. Earlier modeling contained limited number of simulated ejecta particles[12], and detailed landing patterns were not obtained for a precise comparison with the shape of the curved ray. More essentially, abundant other rays are visible in this area (Supplementary Fig. 1), but both the pixel scale and spectral resolution of Mariner 10 images are not adequate to conclude that the curved ray was indeed formed by a single impact.

In this work, using high-resolution and multi-band optical images returned by the MErcury Surface, Space ENvironment, GEochemistry and Ranging (MESSENGER) spacecraft[17], we show that the curved ray was formed by the late-Kuiperian-aged (≪300 Myr[18]) Hokusai crater, which is located at the northern high latitude of Mercury (Fig. 1a). Based on comprehensive modeling for ballistic trajectories of impact ejecta, this study further reveals that the sharp change of orientation was not caused by a poised faster spin speed of Mercury, but due to

[1]Planetary Environmental and Astrobiological Research Laboratory, School of Atmospheric Sciences, Sun Yat-sen University, Zhuhai, China. [2]CAS Center for Excellence in Comparative Planetology, Hefei, China. [3]These authors contributed equally: Rui Xu, Zhiyong Xiao. ✉e-mail: xiaozhiyong@mail.sysu.edu.cn

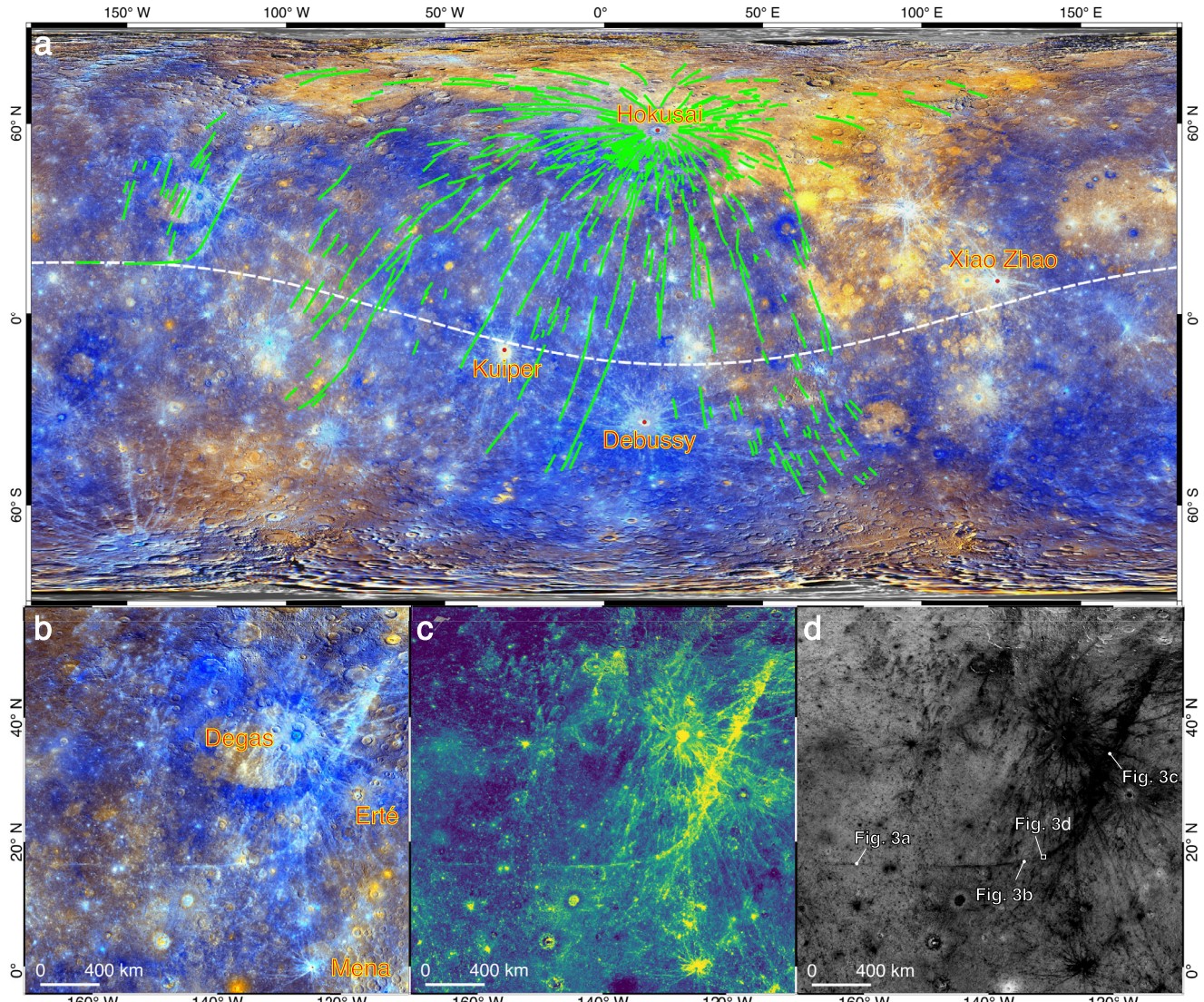

**Fig. 1 | The hyperbola-shaped ray of the Hokusai crater. a** Visible rays of the Hokusai crater (rim-to-rim diameter $D = 114$ km; 16.8°E, 57.8°N) cover most parts of the planet (green lines). The distribution of the rays was updated from an earlier mapping[26] (see Methods). The white dashed curve is a planet-centered great circle that passes the western branch of the curved ray. **b–d** MESSENGER enhanced color mosaic, optical maturity index, and spectral slopes of reflectances at visible to near-infrared (see Methods) for the curved ray, respectively. Image IDs used for this figure are listed in Supplementary Table 1.

azimuthally non-uniform ejection angles. Analyses of cratering mechanics suggest that such homeless distal ejecta are common on planetary surfaces, and they have profound geological implications.

## Results

### The hyperbola-shaped ray formed by the Hokusai crater

This curved ray is more-or-less continuous in extension, and it consists of three branches in terms of orientations (Fig. 1b). The western branch is parallel to the 16°N latitude line and the eastern branch trends northeast to southwest, and they are connected with an intersection angle of about 120° at the orientational inflection branch. Abundant other impact rays exist in this region, and most of them are traced back to the Degas, Erté and Mena craters (Fig. 1b). However, both the western and eastern branches of this curved ray exhibit different orientations than the other rays (Fig. 1b). At visible to near-infrared wavelengths, the entire curved ray exhibits similar optical maturities (Fig. 1c) and slopes of reflectance spectra (Fig. 1d), which are distinctive from those of the other rays in this region. Therefore, this study confirms that the curved ray was most likely formed by a same impact event[12].

The eastern branch of this curved ray was mapped as part of the global ray system of the Hokusai crater (Fig. 1a)[19–21]. However, the western branch cannot be traced radially back to Hokusai (Fig. 1a). The great circle that passes the western branch of this ray does not directly transect rims of impact craters that have obvious rays (Fig. 1a). The Kuiper ($D = 62$ km), Debussy ($D = 81$ km), and Xiao Zhao ($D = 24$ km) craters are visible at not far from this great circle, but visible rays formed by these craters do not extend to the region of the curved ray (Fig. 1a). The Hokusai crater was formed by an oblique impact as evident by its asymmetric rays (Fig. 1a), and the impactor had a trajectory from northeast to southwest, with an inclination angle of about 30–40° from the surface tangent[19,20]. Therefore, the curved ray was likely formed by downrange ejecta excavated by the Hokusai crater, with ballistic ranges of over 4500 km from the crater center (Fig. 1a).

Impact rays with non-radial distributions with respect to their source craters are common on planetary bodies, such as the Moon[8,22,23], Mars[24], and Mercury[20]. However, secondary craters (secondaries) are usually visible in such rays, and their morphology is indicative to the azimuth of landing of the impacting ejecta (Fig. 2). Physical simulations for the formation of secondaries revealed that

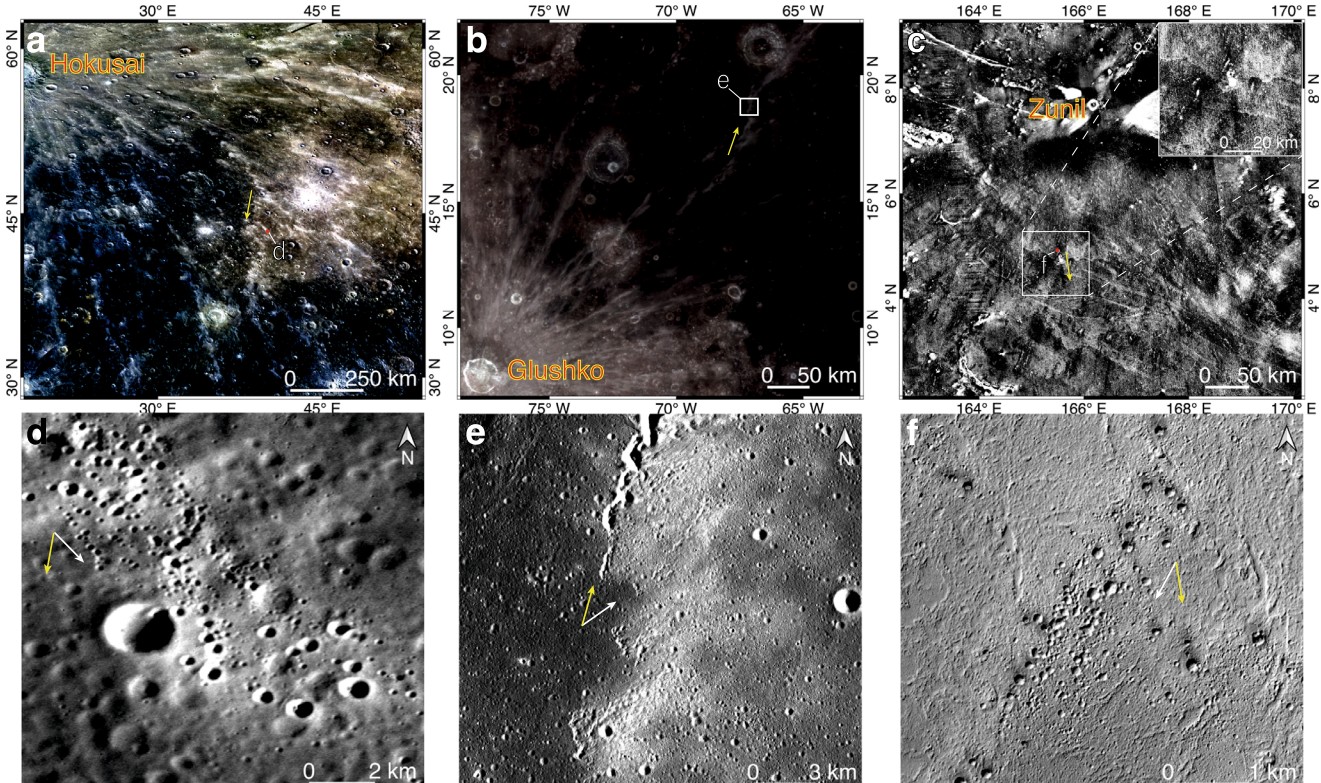

**Fig. 2 | Non-radial rays on planetary bodies that are composed by discrete segments of radial rays.** Yellow arrows denote regional extensions of the non-radial rays. White arrows denote azimuths of landing of ejecta that formed the local ray segments, which are determined based on both the asymmetric rim elevations of secondaries and the deposition directions of their ejecta. **a**, **d** A case of the Hokusai crater on Mercury. **b**, **e** A case of the Glushko crater ($D = 40$ km) on the Moon. **c**, **f** A case of the Zunil crater ($D = 10.3$ km) on Mars, and the inset in **c** shows an enlarged view for the non-radial ray that has a lower thermal inertia than the surroundings. Image IDs used for this figure are listed in Supplementary Table 1.

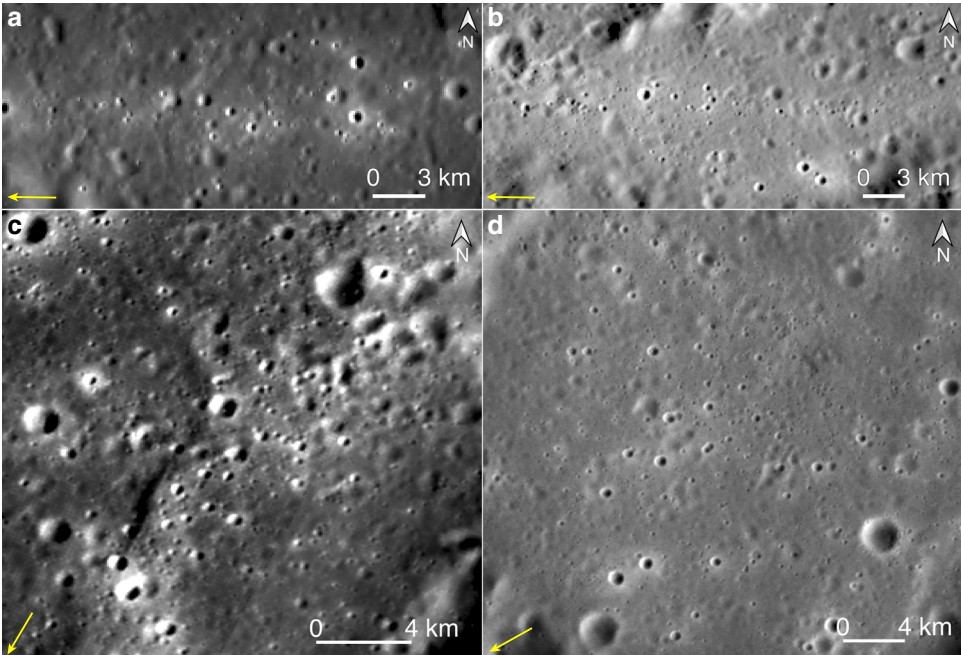

**Fig. 3 | Chains of secondaries in the curved ray on Mercury. a–d** Chains of secondaries in the western, eastern, and orientational inflection branches of the curved ray, respectively. Their locations are marked in Fig. 1d. Yellow arrows denote local extensions of the curved ray. Image IDs used for this figure are listed in Supplementary Table 1.

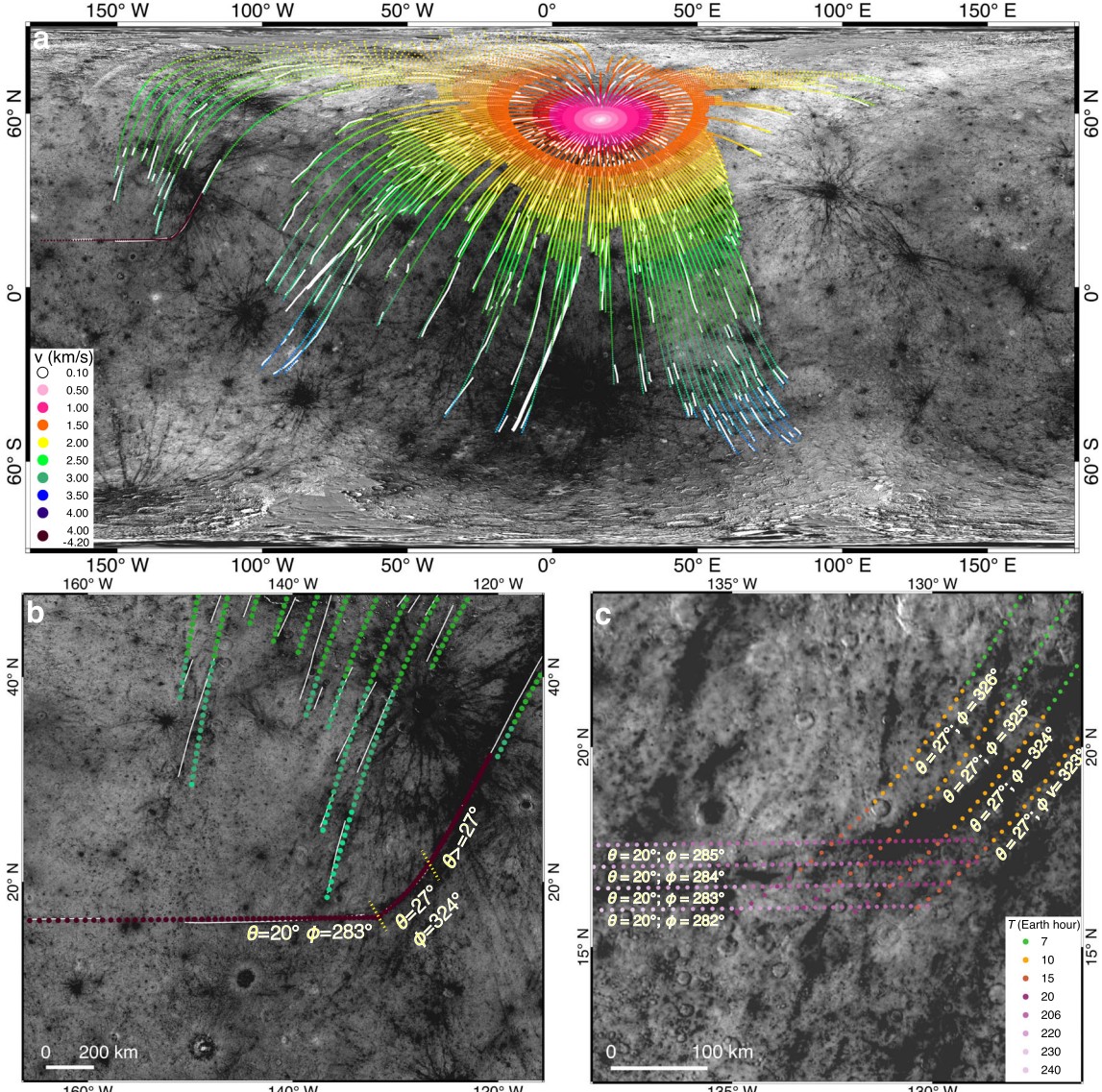

**Fig. 4 | Best-fit ejection conditions for Hokusai's ejecta according to its visible impact rays. a** Best-fit ejection conditions as compared to the global ray system (white lines). Except for the curved ray in question, most rays are consistent with being formed by ejecta with uniform ejection angles of $\theta = 35$–$55°$ (Supplementary Fig. 4). The reported case is based on $\theta = 45°$, excavation from the crater center of Hokusai, and the current rotational period of Mercury. Cases with the other excavation positions are reported in the Supplementary Figs. 3–4. Modeled ejecta particles are color-coded according to their ejection velocities. Bold white lines highlight several other non-radial rays that also indicate abnormal ejection conditions (Supplementary Fig. 5). **b** Enlarged view for the best-fit ejection conditions of ejecta that formed the curved ray. The color bar in panel **a** is also appliable here. **c** Flight times of Hokusai's ejecta (in Earth hours) that can land around the orientational inflection branch of this curved ray. The assumed ejection conditions are annotated besides each group of ejecta that has the same azimuth of the excavation ($\phi$). Image IDs used for this figure are listed in Supplementary Table 1.

during low velocity and oblique impacts by swarms of ejecta fragments, the raised rims of secondaries are more pronounced at the uprange but subdued or even completely destroyed at the downrange of the landing azimuth, and ejecta excavated by secondaries are preferentially deposited towards the downrange[25]. As evident by the morphology of secondaries in such non-radial rays, they were formed by laterally adjacent and/or connected clusters of ejecta that had different azimuths of landing, but each cluster follows a radial distribution with respect to the source crater.

Chains of pristine secondaries that are as large as >1 km in diameters are resolvable in some parts of the curved ray on Mercury, and their orientations are identical with the local ray segments (Fig. 3a–c). Unlike typical secondaries that exhibit irregular shapes (Fig. 2), the secondaries here are circular in shapes and relatively dispersed in the spatial distribution (Fig. 3), indicating that fragments forming

these secondaries were more dispersed in the spatial distribution when landed[25], possibly with rather steep elevation angles with respect to the surface tangent[26]. Feathery high-reflectance ejecta excavated by these secondaries are confined within the local ray segments (Fig. 3a–c), instead of forming spikes that have different orientations (Fig. 2d–e). Therefore, locally along this curved ray, distal ejecta of Hokusai were landed in sequence from a same azimuth[26], such as those formed the western (Fig. 3a–b) and eastern (Fig. 3c) branches of the curved ray, respectively. Abundant pristine secondaries are also visible in the orientational inflection branch of this curved ray (Fig. 3d). Their possible provenances are not clear due to both their circular shapes and the occurrence of many other impact rays that have various orientations here (Fig. 1c). Therefore, for Hokusai's ejecta that formed the western and eastern branches of this curved ray, they were not necessarily with the same azimuth of landing.

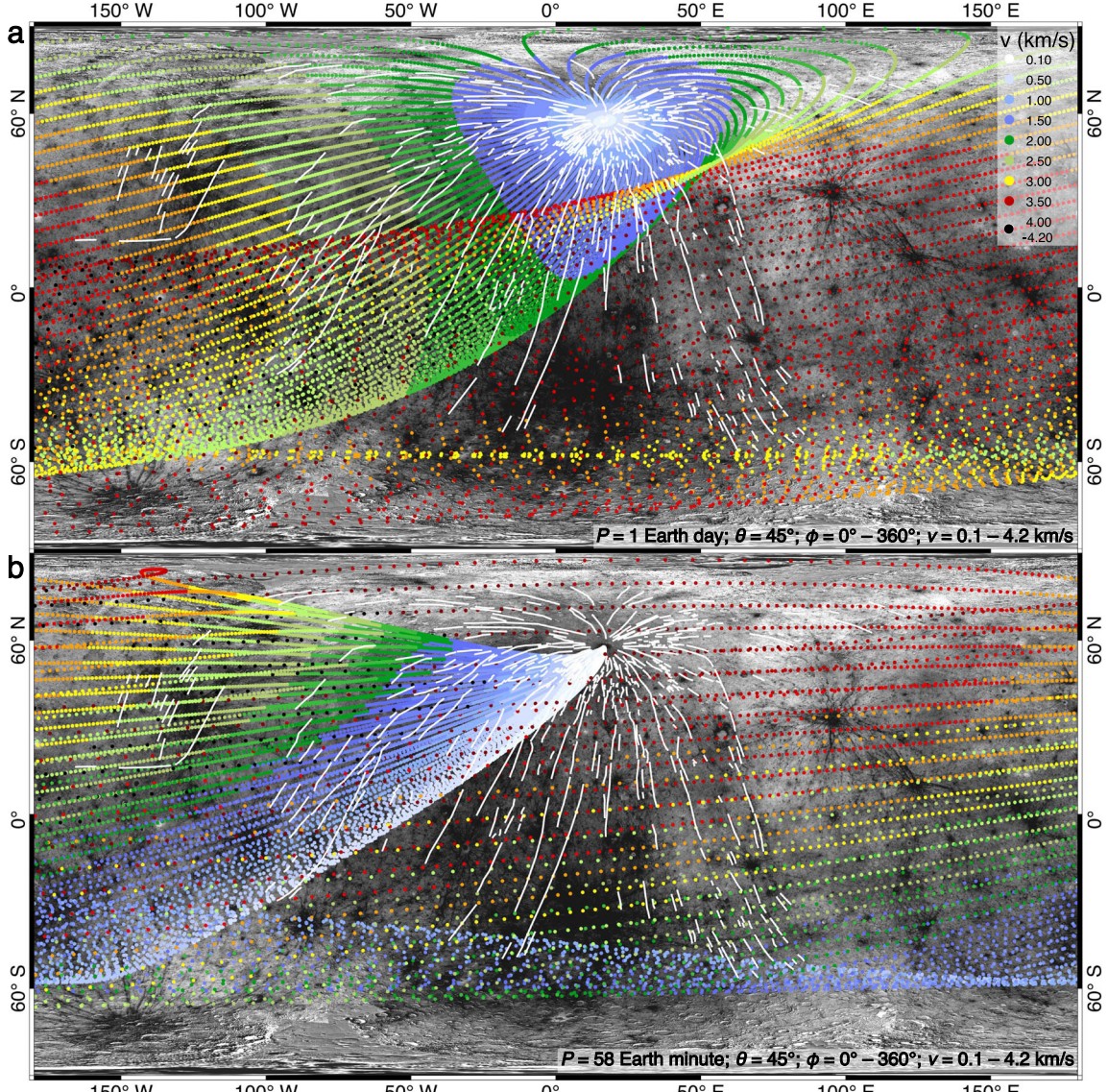

**Fig. 5 | Landing positions of Hokusai's ejecta on a faster rotating Mercury.**
**a**, **b** Cases with assumed rotation periods of Mercury as $P = 1$ Earth day and $P = 58$ Earth minutes, respectively. The modeled ejecta are assumed to be launched from the center of the Hokusai crater, and the landing sites are color-coded according to their ejection velocities ($v$). Image IDs used for this figure are listed in Supplementary Table 1.

## Falsification of a recent faster rotation of Mercury

By updating the ballistic trajectory model for impact ejecta on rotating planetary bodies[12], this work modeled landing positions and flight times ($T$) for large numbers of ejecta particles (see Methods). The model did not include the effect of rotational dissociation[27] or mutual collisions[9,10] of ejecta during flight, which may change their post-ejection trajectories. Therefore, ejecta with the same azimuth of landing had the same azimuth of ejection. The model-predicted landing positions were compared with the shape of the curved ray to deduce the azimuths of excavation ($\phi$; north is zero) from the transient cavity (see Methods), ejection velocities ($v$), ejection angles with respect to the surface normal ($\theta$; vertical is zero), and rotation period of Mercury ($P$). For each secondary crater in the curved ray, multiple combinations of solutions existed for $\phi$, $v$, $\theta$, and $P$, because the only known parameters in the inverse calculation were the observed locations of secondaries and their initial region of excavation from the transient cavity. To narrow down the best-fit solution, the uniform azimuth of ejection of ejecta that formed the secondaries in the western ray branch was referred as a critical constraint (see Methods).

At the earlier-predicted region of impact that formed this curved ray[12], young craters with obvious rays are not observed (Supplementary Fig. 2). Following the assumed rotational period of 5.9 Earth days[12], this study seeded a much larger number of ejecta particles in the model (Supplementary Fig. 2a). At the western branch of the curved ray, distal ejecta launched from the hypothesized source impact region exhibit landing patterns that are inclined towards the south (Supplementary Fig. 2b). Therefore, the curved ray was not formed by the hypothesized low latitude impact on a faster spinning Mercury.

The global ray system of Hokusai, including the eastern branch of this curved ray, is generally consistent with the spatial pattern of landing positions of ejecta that had a uniform $\theta = 45°$ (Fig. 4a). This good match holds for various assumed positions of excavation from the transient cavity of Hokusai (Supplementary 3). Strictly, rays less than about 4000 km from the Hokusai crater are generally consistent with being formed by ejecta with normal ejection angles of about 35° to 55° (Supplementary Fig. 4). This is consistent with the conventional knowledge that ejection angles during planetary impacts are

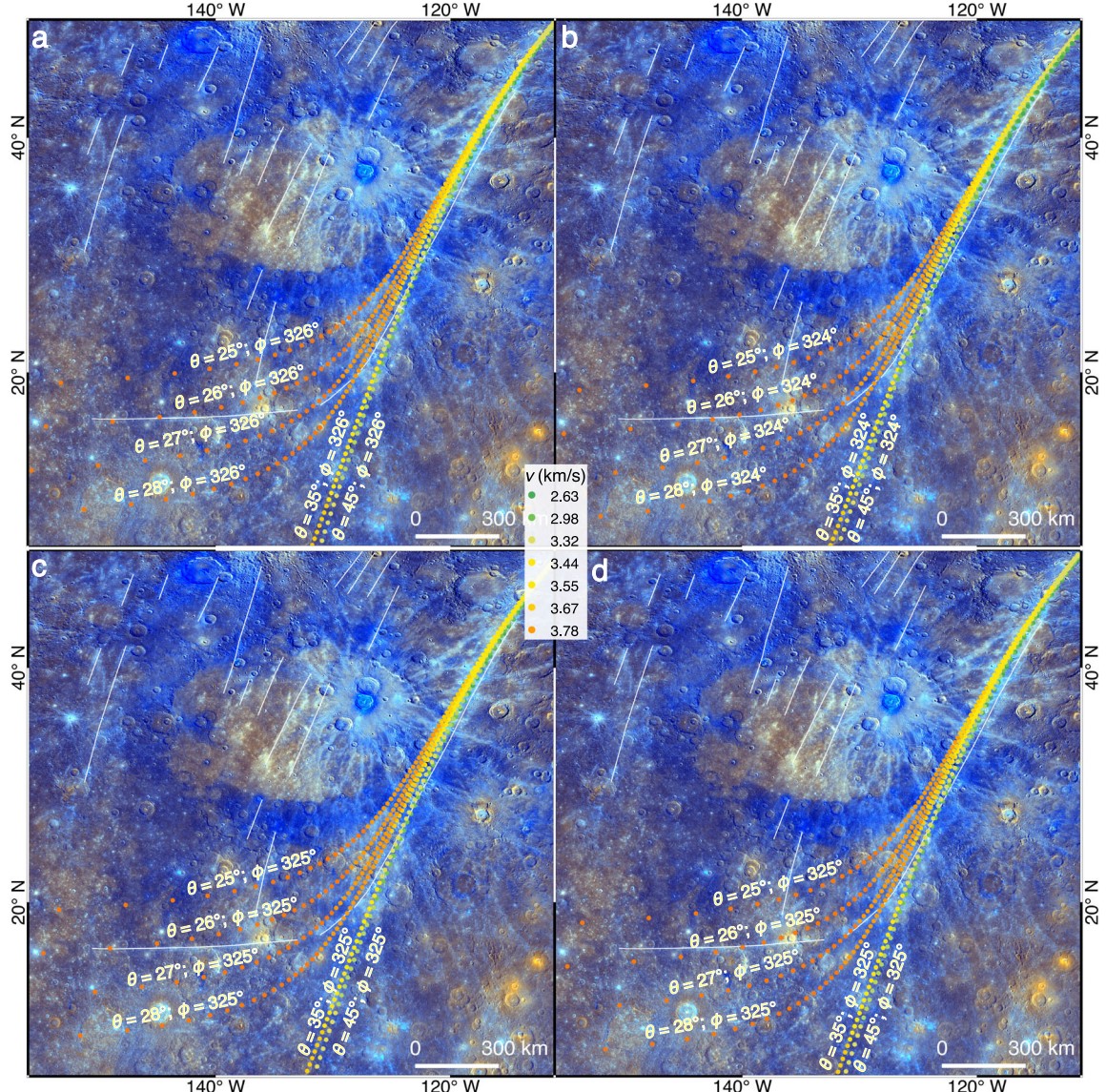

**Fig. 6 | Ejecta with a same azimuth of excavation cannot form the entire curved ray. a–d** Modeled landing sites of ejecta that were excavated from the eastern, western, southern and northern rims of the transient cavity of Hokusai, respectively. For each case, different ejection angles ($\theta$) are tested for ejecta with the same azimuth of excavation ($\phi$). The best-fit $\theta$ and $\phi$ are derived by comparing the landing positions with the eastern and orientational inflection branches of the curved ray. The modeled ejecta particles are color coded according to their ejection velocities ($v$). The rotation period of Mercury ($P$) is the same as it is today. Image IDs used for this figure are listed in Supplementary Table 1.

concentrated around 45° (refs. [3,4]), and most distal ejecta follow more-or-less radial distributions[28,29].

Assuming that the Hokusai crater was formed when Mercury had a faster rotation speed, the larger Coriolis force could systematically shift the predicted landing positions of distal ejecta, preferentially towards the west and higher latitudes (Fig. 5). However, even assuming an unrealistically short rotation period of $P = 58$ Earth minutes, Hokusai's distal ejecta that had a same $\phi$ cannot land along the western branch of this curved ray (Fig. 5b). Furthermore, in such cases, the modeled distribution of landing positions would completely mismatch with the observation (Fig. 5b). Therefore, Mercury did not have a faster rotation speed when the Hokusai crater was formed.

### Azimuthal asymmetry of ejection angles

Hokusai's ejecta that had ejection angles of $\theta = \sim 25–65°$ could be landed along the eastern branch of this curved ray (Figs. 4, 6 and Supplementary Figs. 3–4). Assuming that ejecta forming the eastern and orientational inflection branches of this curved ray had a same $\phi$, the best-fit model predictions yield $\theta = \sim 25–27°$ and $\phi = \sim 324–326°$, depending on the excavation position in the transient cavity (Fig. 6). Therefore, the orientational inflection branch of this curved ray were formed by ejecta with rather steep $\theta$, and those with the same $\phi$ but smaller $v$ and larger $\theta$ could form the eastern branch. On the contrary, the western branch of this curved ray cannot be formed by ejecta with the same $\phi = \sim 324–326°$, regardless of the different assumed $v$, $\theta$ or positions of excavation, because the predicted landing positions would profoundly incline towards the south at these positions (Fig. 6). Therefore, the entire curved ray cannot be formed by ejecta excavated from a same $\phi$.

While ejecta with various combinations of $\phi$ and $\theta$ can land on the western branch of the curved ray (Fig. 6 and Supplementary Figs. 3–4), the single landing azimuth of ejecta that formed the secondaries in the western ray (Fig. 3a–b) places a tight constraint on the required ejection conditions, yielding a best-fit $\phi = \sim 283°$ and $\theta = \sim 20°$ (Fig. 4c). The predicted steep ejection angles are consistent with the circular shapes of secondaries in the ray (Fig. 3a–b). It is notable that ejecta with

$\phi = \sim283°$ and smaller $v$ should be able to form more rays adjacent to the eastern ray branch, which are not visible (Fig. 4a). This observation is consistent with the knowledge that impact rays on planetary surfaces usually appear as discontinuous ray segments[30].

Unlike the western branch of the curved ray, the other rays of Hokusai that are over 4000 km from the crater center cannot be formed by ejecta with $\theta = \sim20°$ (Supplementary Fig. 4a, b). Model fitting yields a minimum $\theta = \sim35°$ for such normal rays (Supplementary Figs. 3–4). On the other hand, the maximum difference of $v$ between the slowest particles that formed the western branch of the curved ray and the fastest particles that formed the eastern branch of the curved ray is only about 150 m/s (Fig. 4b). For comparison, during the entire course of the excavation stage of typical planetary impacts, ejection velocities change from supersonic to zero, but ejection angles have a limited variation of about 15° (refs. [3–5]). Therefore, the ejecta that formed the western branch of this curved ray had abnormally steeper ejection angles than those with similar $v$ but different $\phi$, demonstrating azimuthally abrupt changes of ejection angles during impact cratering.

## Discussion

Ejecta launched during the excavation stage of impact cratering are mainly subsonic in velocities[1], and those with $v > 4$ km/s (Fig. 4b) are highly shocked impact melt[27], which would not form secondaries upon landing[31]. Ejecta formed by impact jetting that occurs in the early contact stage are also an unplausible source, which exhibit ejection velocities comparable to or even larger than the impact velocity[32,33]. Secondaries on planetary bodies are believed to be mainly formed by fragments excavated during impact spallation[1]. Owing to the overlapped arrival times of shock and rarefaction waves at the shallow surface of target materials, spall fragments can have ejection velocities larger than the escape velocity but experience shock levels much lower than melting, typically half of the dynamic crushing strength of silicate rocks[27]. Therefore, the observed secondaries in the curved ray (Fig. 3) were most likely formed by solid spall fragments.

Ejection angles of spall fragments, usually assumed to be 45° (refs. [29,34]), are the vector sum of particle velocities induced by shock and rarefaction waves[27]. Recent high-resolution numerical simulations are capable to capture the evolution of ejection angles and velocities during jetting and early spallation in planetary impacts[33,35]. Spall fragments excavated by oblique impacts exhibit azimuth asymmetry in the ejection velocities and angles[33]. Those launched toward the downrange have shallower ejection angles ($\theta > 45°$) but larger ejection velocities (up to about 2 times the impactor velocity). Those launched towards the uprange exhibit rather steep ejection angles of $\theta = \sim10°$ but substantially smaller ejection velocities[33]. The curved ray on Mercury reflects abrupt changes of $\theta$ at certain $v$ and $\phi$, instead of systematical variations of ejection conditions[33,36]. Therefore, the oblique impact that formed the Hokusai crater was not an obvious reason to form the curved ray.

Propagation paths of shock and rarefaction waves are affected by non-enumerable issues that could introduce heterogeneities in the shock impedances and/or geometries of shocked materials[27]. Recent numerical simulations[6,37] and laboratory impact experiments[37] revealed pronounced effects of topographic undulations on the ejection angle of spall fragments. Close to topographic undulations of target surfaces that have appropriate geometries, locally steepened ejection angles were observed[7], mainly caused by local changes of propagation paths of shock and rarefaction waves. The Hokusai crater was formed in the northern volcanic plains, where scattered topographic and structural discontinuities were abundant, such as wrinkle ridges, unflooded crater rims, and secondaries formed by earlier craters (Fig. 7a)[38]. Therefore, before the impact event that formed the Hokusai crater, abundant conditions that could cause local variations of propagating paths of shock and rarefaction waves existed in the

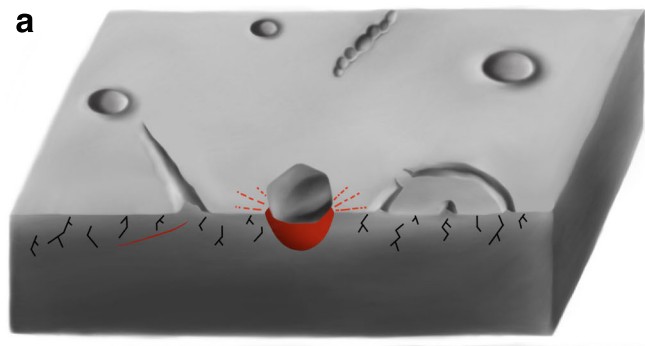

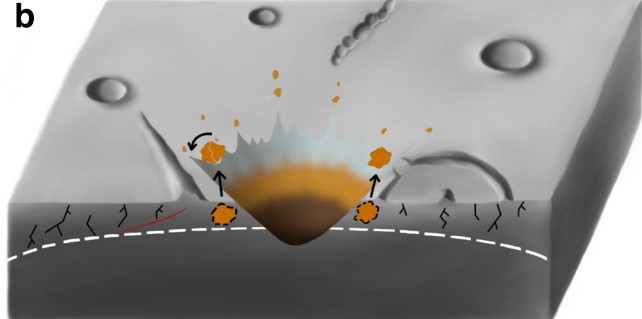

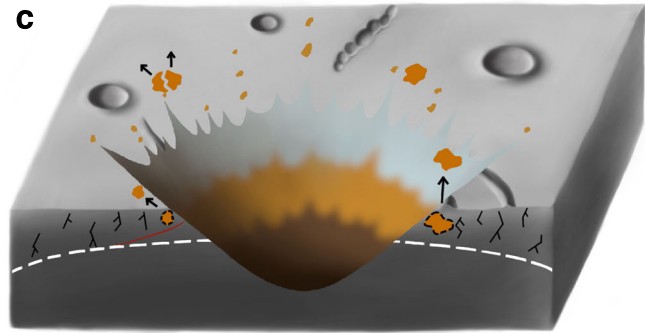

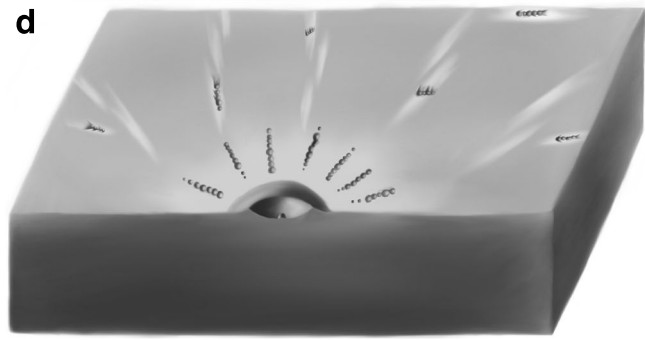

**Fig. 7 | Illustration for the possible cause of steep ejection angles during impact spallation that formed the Hokusai crater. a** Initial contact between an irregular-shaped projectile and the northern volcanic plains on Mercury. Subsurface fractures and various forms of topographic undulations existed in the pre-impact target, such as buried impact craters, wrinkle ridges, and earlier-formed secondaries. **b, c** Impact spallation in the near-surface target materials. Spall fragments excavated from the vicinity of topography undulations may exhibit abrupt changes of ejection angles. Rotational segmentation may occur in some fast-spinning spall fragments (black curved arrow in **b**). **d** Impact rays and their interior secondaries. Distal ejecta with abnormal ejection angles would form non-radial distributions of secondaries in distal rays. The azimuth of landing of impact ejecta can be derived based on the morphology of secondaries.

pre-impact target (Fig. 7b, c), permitting abrupt changes of $\theta$ to form the curved ray (Fig. 7d).

Several other observations support that abrupt changes of $\theta$ are not rare during impact spallation, although such ejecta are minor

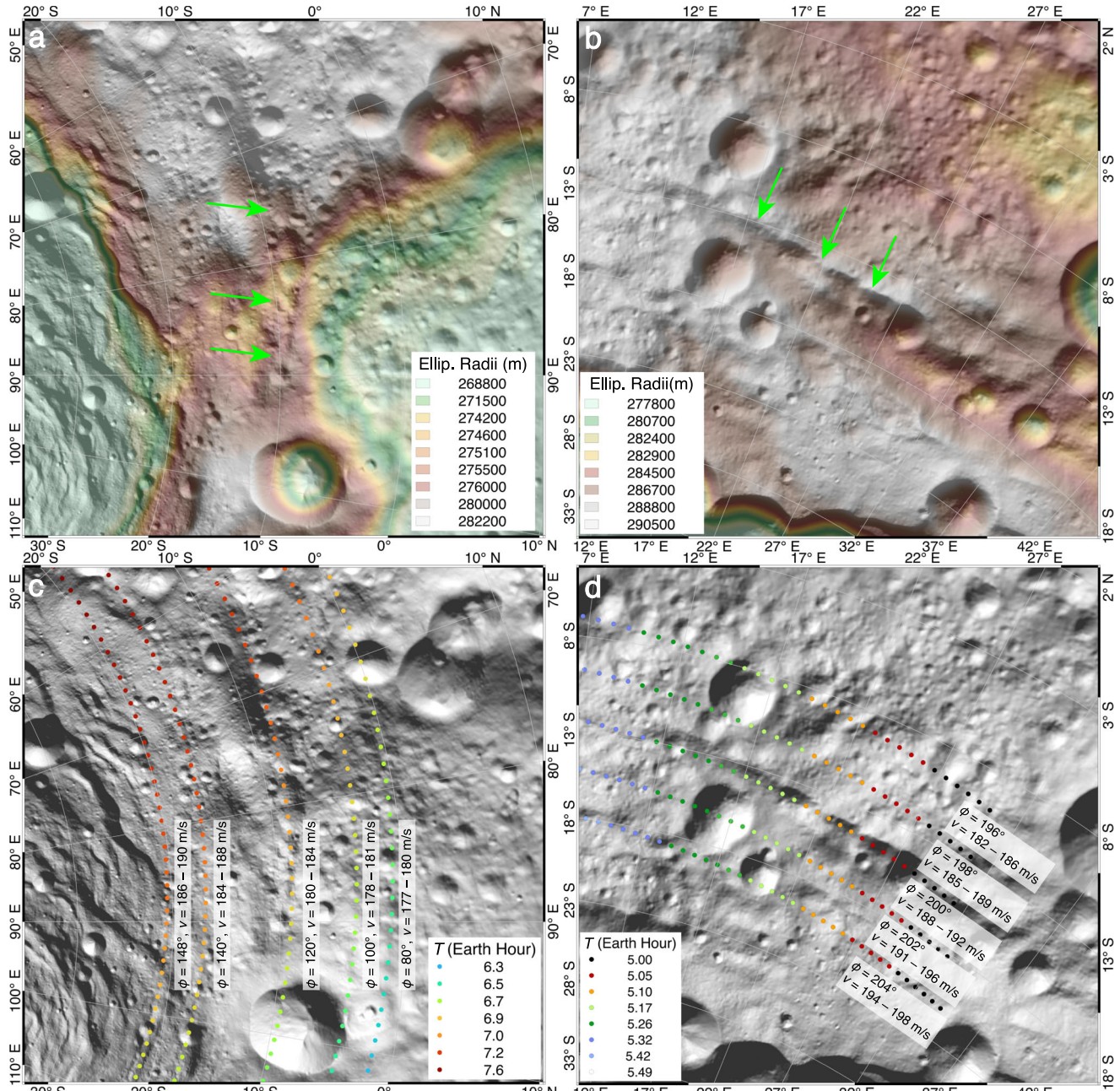

**Fig. 8 | Some circumferential troughs on walls and rims of the Rheasilvia basin at the south pole of 4Vesta may be formed by secondary impacts. a, b** Many of the troughs occur as pit chains that have raised rims (green arrows). **c, d** Predicted landing positions of ejecta from the Rheasilvia basin can occur along the roughs. Ejection conditions in the model are annotated in the legend. Results for all the modeled particles are shown in the Supplementary Fig. 6. Image IDs used for this figure are listed in Supplementary Table 1.

compared to the overall radial distributions of impact ejecta. On melt veneers that swept over the continuous ejecta deposits of the Hokusai crater, self-secondaries were discovered based on over-lapping relationships[19]. Estimations of ejection conditions suggested that the self-secondaries were formed by spall fragments that had $v > -1$ km/s and $\theta < 10°$ (ref. [19]). For comparison, most spall fragments with similar $v$ but $\theta > 25°$ formed radial rays (Fig. 4a) and their interior secondaries[26]. It is notable that the observed self-secondaries of Hokusai have a slightly larger spatial density at the uprange con-tinuous ejecta deposits[26], consistent with the averagely steeper $\theta$ of spall fragments launched towards the uprange as revealed by impact simulations[33]. Furthermore, among the other non-radial rays of Hokusai, some contain chains of secondaries that were landed along

the local ray segments (Supplementary Fig. 5). Trajectory modeling suggests that compared to their adjacent radial rays, such non-radial rays may be also formed by ejecta with abnormally large or small ejection angles (Supplementary Fig. 4). Correlated morphological study for secondaries in such rays could reveal their azimuths of landing, suggesting that about 6% of Hokusai's rays are non-radial in distributions and formed by ejecta with abnormal ejection angles (e.g., bold white lines in Fig. 4a). Considering that heterogeneity is inherent to physical and chemical properties of planetary materials, abrupt changes of ejection angles should be common during impact spallation, so that non-radial distributions of impact ejecta should not be rare on planetary surfaces. This prediction explains the recent discovery of common existences of self-secondaries on rims of lunar

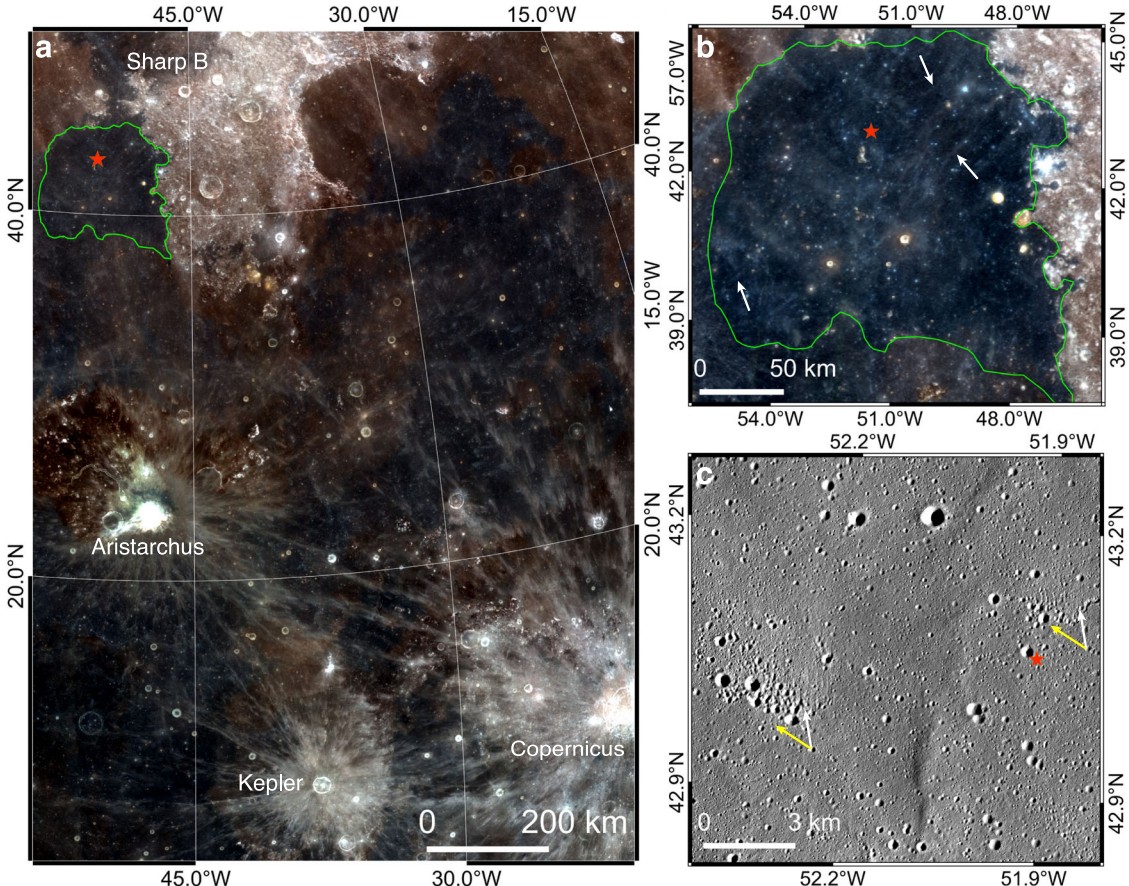

**Fig. 9 | Non-radial extension of chains of secondaries near the Chang'E-5 sampling site (red star). a** Regional context of the Chang'E-5 sampling site. **b** Faint impact rays (white arrows) are widespread in the sampling region. **c** Chains and clusters of secondaries near the sampling site. Their lateral orientations (yellow arrows) are not radial to the Aristarchus crater[49], but the morphology of the secondaries indicates that the source ejecta had an azimuth of landing (white arrows) that is radial to the Aristarchus crater. Green curves circle the mare unit where Chang'E-5 was landed[49]. Image IDs used for this figure are listed in Supplementary Table 1.

simple craters, which also required abnormally steep ejection angles during impact spallation[39].

While distal ejecta with non-radial distributions are best manifested by immature rays, upon aging, their source craters would be unobvious, possessing difficulties to interpret the formation of planetary landscapes. On 4Vesta, the rims and walls of the Rheasilvia basin are largely encircled by circumferential roughs, such as the Divalia Fossae (Supplementary Fig. 6a). The troughs were believed to be tectonic features formed later than the Rheasilvia basin, as either fault-controlled graben[40] or opening-mode cracks[41]. However, many of the troughs appear as closely aligned pit chains that have raised rims (Fig. 8a, b), which are not different from typical chains of secondaries on planetary surfaces[42]. Spall fragments excavated by the Rheasilvia basin could have flight times longer than the formation time of this basin ($T > 5\,h$[43]; Fig. 8c, d) due to the small surface gravity (0.025 $g$). With a fast rotation speed of 4Vesta, ejecta with various ejection conditions could form curved chains of secondaries along the throughs (Fig. 8c, d; Supplementary Fig. 6), providing an alternative interpretation for their origin as secondary impacts rather than tectonism.

On the Moon, the Chang'E-5 mission recently returned regolith from one of the youngest mare units[44]. Minor exotic materials that were ejected from the other regions of the Moon are discovered in the regolith, and their provenances are valuable to deduce the lithological diversity of the Moon and the evolution of local regolith[45–47]. Faint impact rays are widespread in the sampling region, yielding ambiguous potential source craters

(Fig. 9a, b). Orientations of chains of secondaries in this region were referred to deduce their source craters[48,49]. Nearby large craters such as Aristarchus and Copernicus (Fig. 9a) were predicted to be major sources of exotic materials in the sampled regolith[50], but they were precluded as possible candidates based on the laterally non-radial extension of secondaries near the sampling site[49] (Fig. 9c). Although these secondaries are not associated with obvious rays, their asymmetrically-raised rims (Fig. 9c) indicate that the source ejecta had a radial azimuth of landing from the Aristarchus crater (Fig. 9a). This observation is consistent with sample analyses that discovered possible ejecta excavated by the Aristarchus crater in the Chang'E-5 regolith[45].

Modeling also reveals that distal ejecta excavated by a same crater but with different $v$, $\theta$ and $\phi$ can land at the same location but at different times and from different azimuths (Fig. 4c), further complicating the deduction of provenances for planetary surface materials and age estimations using crater statistics. On Mars, overlapped distal secondaries that were formed by a same crater also exist (Supplementary Fig. 7a). Figure 10a, b show the intersection region between a curved ray and a radial ray of secondaries, which were both formed by the Corinto crater[51]. Modeling reveal that the source ejecta of the secondaries may be excavated with various ejection conditions, and their landing occurred at different times and from different azimuths (Supplementary Fig. 7b, c). Unlike typical secondaries that have irregular shapes and clustered spatial distributions, many secondaries on Mars are more circular in shapes and dispersed in

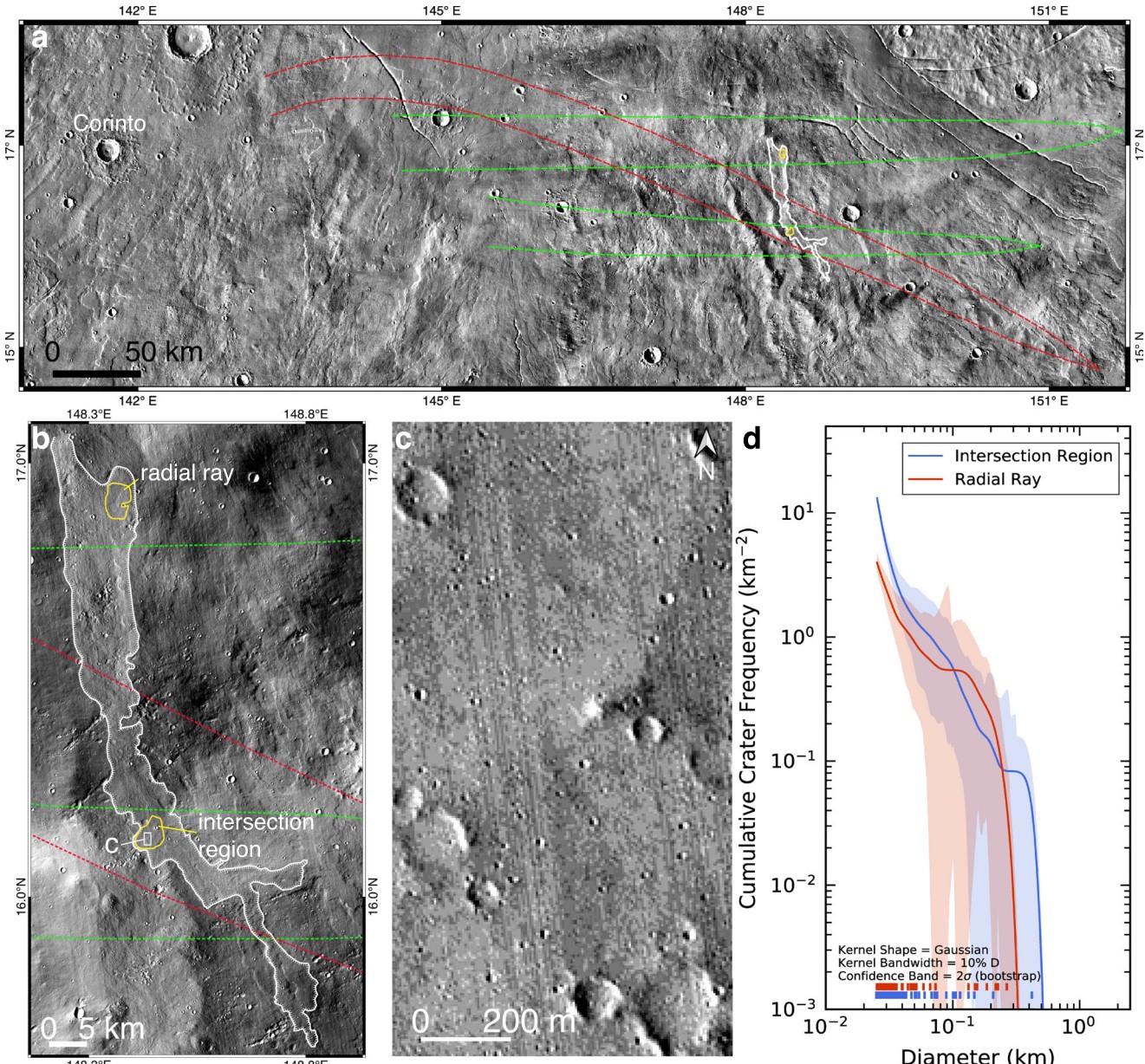

**Fig. 10 | Distal ejecta formed by a same crater may exhibit overlapping relationship at a same site. a, b** The Corinto crater on Mars ($D = 13.7$ km) formed both radial (green curves) and non-radial (red curves) distributions of secondaries, and some of them cross the same region (Supplementary Fig. 7a)[51]. A long lava flow (white curves) crosses the intersection region of two rays of secondaries formed by the Corinto crater. The yellow polygons are two counting areas on the lava flow, where crater densities are measured and compared. **c** Morphology of impact craters in the intersection region of the two rays of secondaries. **d** The lava flow exhibits a much larger density of impact craters ($D = -25–45$ m) in the intersection region than that in a single radial ray. The cumulative crater size-frequency distributions are plotted using a probability distribution function, and the error bars are derived based on bootstrap estimations[66]. Image IDs used for this figure are listed in Supplementary Table 1.

the spatial distribution[52] (Fig. 10c), so they are not easily distinguishable from same-sized primary craters. On a same lava flow that crosses the intersection region (Fig. 10a, b), crater statistics that excluded obvious secondaries yielded a much higher density in the intersection region, inconsistent with the same age of the lava flow (Fig. 10d). Owing to non-radial distributions of distal ejecta, such mendaciously large densities of impact craters should be common on planetary bodies, especially those host abundant circular secondaries, such as Mars and Mercury[26]. This phenomenon would be even more pronounced on faster rotating bodies that have young surfaces, such as Europa (Supplementary Fig. 8). Therefore, due to contaminations by scattered secondaries that had non-radial distributions, larger densities of impact

craters may not be strictly corresponded to larger model ages[53], reiterating the relatively large uncertainty in the age determination using small impact craters[54].

## Methods

### Processing and usage of remote sensing data

For Mercury, images obtained by both the 750 nm band narrow angle camera (NAC) and the multiple-band wide angle camera (WAC) of the Mercury Dual Imaging System[55] onboard the MESSENGER spacecraft are used in this study. Various versions of MDIS WAC global mosaics are used (all with a pixel scale of 665 meters), mainly to show the morphology and spatial distribution of impact rays. These include (1) a false color global mosaic that has red, green and blue bands as

reflectances at 1000, 750 and 430 nm, respectively; (2) an enhanced global color mosaic of Mercury that is derived from the global eight-color mosaics[56] based on a principal component (PC) analysis, and PC2, PC1 and the ratio of reflectances at 430 nm wavelength to that of 1000 nm are used as the red, green and blue channels, respectively; (3) a global map of slopes of reflectance spectra at visible to near-infrared wavelengths, and each pixel is the ratio between reflectances at 1000 and 430 nm wavelengths[57]; and (4) a global map of optical maturity of Mercury, and each pixel is the ratio between the average reflectance of the eight-band MDIS WAC spectrum and the spectral slope mosaic[57]. In addition, single frames of MDIS NAC images are used to study the detailed morphology of secondaries inside impact rays.

For the Moon, multiband images obtained by the Lunar Reconnaissance Orbiter Camera[58] are used to investigate the lateral extension of impact rays. High-resolution monochrome images obtained by the Terrain Camera onboard the SELENE-Kaguya spacecraft are used to investigate the morphology of secondaries in impact rays.

For Mars, both the daytime and nighttime global mosaics obtained by the Thermal Emission Imaging System[59], Mars Odyssey, are used to investigate the lateral extension of impact rays. Images obtained by the Context Camera onboard the Mars Reconnaissance Orbiter[60] are used to investigate the morphology of secondaries in impact rays.

For 4Vesta, the global digital elevation model constructed using data obtained by the Dawn mission and its shade relief models[61] are used to investigate the morphology of the circumferential troughs that are developed around the Rheasilvia basin.

For Europa, the global monochrome mosaic obtained by the Voyager and Galileo spacecrafts are used[62].

All the imagery and topography data used in this study are available in the public domain, and their IDs and available addresses are in the Supplementary Table 1.

### Updating the ray system of Hokusai

Impact rays formed by the Hokusai crater on Mercury were mapped before based on the false color mosaic and enhanced global color mosaics (see Methods) of Mercury[19,26]. The main criteria used in the manual mapping were the radial distribution of rays that are obviously sourced from the Hokusai crater and the visual similarities of reflectances of rays along a given azimuth. This work follows the same criteria when updating the ray system of Hokusai. To assist the mapping, the global spectral slope map and optical maturity index map (see Methods) are used additional to the other datasets. The two new maps have advantages in stressing reflectance contrasts of rays that were formed by different craters (Fig. 1c, d). It is notable that the mapping was a subjective process, and the ray system of Hokusai may be further modified with additional datasets (e.g., thermal-infrared data and radar data) or different mapping perspectives[20].

### Trajectory modeling of impact ejecta on airless planetary bodies

Forward modeling for landing positions of impact ejecta is performed to investigate the required ejection conditions of ejecta that formed secondaries in distal impact rays. Model inputs include the initial ejection velocity ($v$; less than the escape velocity of the target body), ejection angle ($\theta$) with respect to the surface normal ($\theta = 0°$ is surface normal and $\theta = 90°$ is surface tangent), ejection azimuth ($\phi$) with respect to the local north ($\theta = 180°$ is local south and $\theta = 270°$ is local west), and ejection position (colatitude $\psi$ and longitude $\lambda$) in the transient cavity.

Large numbers of ejecta particles are seeded in each simulation, typically over 10,000, according to assigned interval of $v$, $\theta$, and $\phi$ (details in https://doi.org/10.6084/m9.figshare.20722891.v4[63]). For each input parameter, the best-fit value was derived based on comparison with the observed distribution

patterns of distal ejecta. The effect of local topography on landing positions and processes that may change post-ejection trajectories of particles, such as mutual collisions and rotational separation during flight, are not included in the model.

The effect of planetary rotation on ballistic trajectories of impact ejecta is considered in the model, and $w$ is rotation angular velocity, $r$ is the radius of the target body. For each simulated particle, the inertial speed of planetary rotation at the ejection position, $b = wr\sin\psi$, is added to the homodromous component of ejection velocity, i.e., eastward for Mercury, 4Vesta and Europa. The new $v_0$, $\theta_0$, $\phi_0$ relative to the sidereal frame are then derived with a simple trigonometry:

$$v_0 = (v^2 + b^2 + 2vb\sin\theta\sin\phi)^{1/2} \tag{1}$$

$$\theta_0 = \arccos(v/v_0\cos\theta) \tag{2}$$

$$\phi_0 = \arccos(\tan\theta/\tan\theta_0\cos\phi) \tag{3}$$

To derive the landing position of the ejecta, the conservation of energy in terms of the *vis-viva* equation, $v_0^2 = GM(2/r - 1/a)$ where $G$ and $M$ are the gravitational constant and target mass, respectively, is used to obtain the semimajor axis $a$ of the trajectory ellipse:

$$a = (2/r - v_0^2/(GM))^{-1} \tag{4}$$

Based on the conservation of angular momentum and ignoring the minor ejecta mass as compared to $M$, the specific relative angular momentum is calculated as $a(1-e^2)G(M+m))^{1/2}$, which equals $v_0 r\sin\theta_0$. Afterward, the ellipse eccentricity ($e$) is determined as:

$$e = (1 - (v_0 r\sin\theta_0)^2/(aGM))^{1/2} \tag{5}$$

The ballistic range of ejecta ($\Delta$) is then calculated based on the ellipse equation $r = a(1-e^2)/(1+e\cos v)$, where v is the true anomaly at the beginning of the trajectory.

$$\Delta = 2(\pi - v) = 2\arccos(1/e - a(1 - e^2)/(re)) \tag{6}$$

Landing positions (colatitude $\psi'$ and longitude $\lambda'$) of ejecta are predicted following ref. [12], and the flight time ($T$) is calculated as $T = 2(\pi\text{-}E + e\sin E)a^{3/2}$, where $E = \arccos(v_0^2/e\text{-}1/e)$.

$$\psi' = \arccos(\cos\psi\cos\Delta + \sin\psi\sin\Delta\cos\phi_0) \tag{7}$$

$$\lambda' = \delta - wT \tag{8}$$

where $\delta = \arccos((\cos\Delta\text{-}\cos\psi\cos\psi')/(\sin\psi\sin\psi'))$, and $\delta$ is considered negative if $\phi_0$ is between $\pi$ and $2\pi$.

### Observational constraints to derive best-fit ejection conditions

Impact rays, especially those closer to the source crater (e.g., rays less than 4000 km from Hokusai), generally follow radial distributions. Ejection conditions of their source ejecta are generally consistent with the canonical knowledge of impact ejection[1], i.e., uniform ejection angles of about $45 \pm 15°$, ejection position from the center of the parent crater, all ejection azimuths around the parent crater, gradual declining of ejection velocities from the escape velocity. For any observed impact ray, non-enumerable combinations of ejection conditions can be derived for ejecta that can land on the ray. To put a tighter constraint on ejection conditions, this work focuses on impact rays that were formed by distal ejecta with a same azimuth of landing. The morphology of

secondaries in rays, especially the spatial extensions of their high-reflectance ejecta and locations of their uprange rims that feature higher topography[25], is used as the reference to evaluate whether or not a ray segment was formed by ejecta with a same landing azimuth. Assuming that the ballistic trajectory of particles is only controlled by their initial ejection conditions when being excavated, a same azimuth of landing means a same azimuth of ejection during impact excavation from the transient crater.

### Ranges of ejection locations from the transient crater of Hokusai

The effect of different positions of ejection from a transient on ballistic trajectories of impact ejecta are considered in this work. The Hokusai crater on Mercury has a rim-to-rim diameter of $D = 114$ km, and the empirical relationship between $D$ and the diameter of the transient crater $D_t$[64] is used to estimate the diameter ($D_t$) for its transient crater.

$$D = 1.4 \times D_t^{1.18}/D_0^{0.18}, if\ D_t \geq D_0/1.3 \quad (9)$$

where $D_O$ is the transition diameter from simple to complex craters, which is about 11.7 km on Mercury[65]. Therefore, the transient crater during the formation of Hokusai is $D_t \approx 61$ km.

## Data availability

All imagery data used in this study are in the public domain. The images of the Moon, Mars and monochrome and color images of Mercury are available at the Planetary Data System Geosciences Node (https://pds-geosciences.wustl.edu). The topography data of 4Vesta and imagery data of Eruopa are available from the Planetary Data System Small Bodies Node (https://pds-smallbodies.astro.umd.edu). The model source data (predicted landing positions of ejecta particles for each simulation done in this work) and GIS shapefiles (e.g., updated rays of Hokusai, crater statistics areas and craters) are available in the Figshare database (https://doi.org/10.6084/m9.figshare.20722891.v4[63]).

## Code availability

The code used to model ballistic trajectories of impact ejecta on airless planetary bodies is available in the Figshare database (https://doi.org/10.6084/m9.figshare.20722891.v4[63]).

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

## Acknowledgements

This research was supported by the National Natural Science Foundation of China under Grant No. 42241108 and 42273040 (Z.X.), the Chinese Academy of Sciences through the B-Type Strategic Priority Program under Grant No. XDB41000000 (Z.X.), the pre-research Project on Civil Aerospace Technologies under Grant No. D020201 (Z.X.), and the China Manned Space Engineering Program.

## Author contributions

R.X. developed the trajectory model and performed data analyses. Z.X. conceptualized the research, designed the methodology, wrote the manuscript, acquired the research fund, and supervised this study. F.L. drafted the schematic diagram, Y.W. joined discussions. J.C. joined discussion and manuscript revision.

## Competing interests

The authors declare no competing interests.
