## [Peer Review File · Nature Communications]

Untrackable distal ejecta on planetary surfacesREVIEWER COMMENTS

Reviewer #1 (Remarks to the Author):

Review for NCOMMS-22-38439

Untrackable distal ejecta on planetary surfaces

Written by Rui Xu and Zhiyong Xiao, Fanglu Luo, Yichen Wang

This manuscript discusses about the origin of non-trackable distal ejecta on Mercury, Vesta and Europa. The authors demonstrated that the variation of the ejection angle during impact ejection can explain the “curved ray”, which is one of the outstanding problems on the geology on Mercury. The authors also ruled out two attractive hypotheses on the origin of the curved ray, which are 1) fast-spinning Mercury at the past and 2) unknown source crater by the detailed surface observation on Mercury and trajectory modeling. The authors proposed that relatively high-ejection angle can be explained by the process called impact spallation. Finally, the authors concluded that distal ejecta launched from single crater would produce complex relationships which tend to lead wrong stratigraphic interpretations.

The manuscript is not well written and not well organized. The presentation quality of the manuscript is poor especially for inter-discipline journals, such as Nature Communications (See major comment 1 and 2). The reviewer, however, agrees with the authors that the finding is important to accurately interpret the chronology of the planetary surfaces. In addition, the reviewer does not agree with the author’s statement that the relatively high ejection angles come from the local topography around impact site (See major comment 3). Consequently, the reviewer supports the publication of the manuscript, but after a major revision.

The reviewer also made comments on the word file.

There are five major comments as follows. Minor comments are made in the attached files.

Major comment 1

The organization of the manuscript is not good. The current manuscript has been hard to understand, at least for the reviewer, what the authors would like to state. The reviewer had to repetitively read the manuscript to understand the content. The reviewer strongly recommends to largely revise the organization of the manuscript to enhance the values of the author’s findings.

The source of the confusions (at least for the reviewer) is the lack of a clear separation between what can be explained by existing knowledge and what is revealed in this study. The former is the radial ray

system on Mercury, Vesta and Europa, and the latter is the origin of the curved ray on Mercury, the origin of troughs on Vesta, and the origin of groove-like structure on Europa. In addition, the authors mentioned about the observational fact and the prediction from impact physics without separation.

The reviewer proposes a better organization, for example, as follows.

Introduction about distal ejecta

- Radial ray system
- There are non-trackable ejecta deposits on Mercury, Vesta, and Europa

High-spin rate has been proposed in previous studies

Model results

- Radial crater rays can be explained by the conventional knowledge about impact cratering (~45 degree of ejection angle)
- The test of the high-spin rate hypothesis.

It can explain the non-radial ejecta pattern on Europa, but cannot explain the curved ray on Mercury.

- High-ejection angle can explain the deposition of the curved ray.

Of course, the authors do not necessarily follow above completely, but the current manuscript is difficult to read because it is just a list of case studies for each object. A drastic change in the structure of the paper may dramatically improve the readability of the manuscript.

Major comment 2

It has been hard to interpret the current figures at least for the reviewer. Although in some respects it is inevitable that the figures have many information, it seems to be necessary to modify it. The reviewer recommends to revise the figures to be that can be read at single glance, or to add supplementary figures. The reviewer made specific comments in the word file. Please examine the separated file in detail.

Major comment 3

The authors stated "Recent numerical simulations revealed pronounced effect of stochastic target topography undulations on spallation angles, and interaction between shock waves and local

topography causes locally steepened spallation angles. It is educible that other forms of structural discontinuities with appropriate geometries, such as lithological boundaries, wrinkle ridges, and/or unflooded crater rims in the northern volcanic plains of Mercury where Hokusai was formed, could change local propagation paths of shock and rarefaction waves, causing abrupt changes of spallation angles (Fig. 3).". The reviewer believes that the relatively-high ejection angles can be explained solely from an impact ejection during oblique impacts. Okamoto et al. (2020), *JGR-Planets*, 125, e2019JE005943 conducted both hypervelocity oblique-impact experiments and shock physics modeling and demonstrated that the ejection angle depends on azimuth angle (See Figs. 11 and 12 of Okamoto et al., 2020). Combined with the author's model, the west–east branch of curved ray on Mercury might provide the information of the impact direction of the impactor, which produced Hokusai crater. The reviewer thinks that the above possibility is worthwhile to add the discussion section to enhance the value of the manuscript.

Major comment 4

Kadono et al. (2020), which has been referred in the manuscript mentioned that the crater rays formed as a consequence of inelastic collision between ejecta particles in an ejecta curtain, and that a mesh pattern between adjoining rays occurs during the growth of ejecta (Fig.2 in Kadono et al.). Nakazawa et al. (2021), *The Planetary Science Journal*, 2:237, further explored the formation mechanism with both impact experiments and N-body simulations. They demonstrated that the mesh pattern formation can be explained by the clustering process of ejecta particles and pattern stretching during ballistic flight would form crater-ray structure. The reviewer is wondering that the non-radial crater rays investigated by the authors might be relics of the mesh pattern. The authors should address such another possibility because the formation of crater rays proceeds via mesh pattern formation according to the previous studies (Kadono et al., and Nakazawa et al.).

Major comment 5

The undefined term “background secondaries” was used repetitively throughout the manuscript.

In addition, it is not clear throughout the manuscript what this term refers to. The authors should explicitly define the term in abstract if this term is so important.

Reviewer #2 (Remarks to the Author):

see attached file

Review of “Untrackable distal ejecta on planetary surfaces” by Xu et al.

In this study, the authors apply ejecta modelling to investigate deflection from radial trajectory of secondary crater rays observed on Mercury. The authors evaluated the influence of debris excavation location with respect of crater centroid, ejection angle and speed as well as the rotational speed of the planet. They show that (1) a major curved ray observed on the surface of Mercury cannot be formed by any other impact craters than Hokusai, and (2) that abrupt changes in ejection angle of spall debris can explain the morphology of curved rays, deviating from radial trajectory. Thus, this study rules out the effect of the Coriolis effect as the main process forming such secondary rays, even on fast rotating bodies.

Overall, I found this paper well written and presented (although this could be improved in some aspects). Modelling and parameters investigated are appropriate, and the main conclusions are supported by the data. Topic is original and outputs have strong implications in the understanding of cratering process, planetary surface evolution and dating, and material displacement from impact cratering.

However, I have a few major points I would like to see addressed by the authors before the study gets published.

Major comments:

- 1- L.63-66: “Analyses of cratering mechanics predict that such homeless distal ejecta should be common on planetary surfaces, supporting a widespread occurrence of background secondaries and hampering reliable stratigraphic interpretations based on crosscutting relationship of distal ejecta.”

If homeless distal ejecta should be common on planetary surfaces, why we do not see such hyperbolic rays associated with the largest and youngest craters formed on Mars (Lagain et al., 2021, <https://hive.curtin.edu.au/research/CDA-94M-release/>)? At least 19 craters associated with secondary crater rays are presented in this work, with rays extending hundreds to thousands kilometres away from their crater of origin. None of them present significant deflection from radial trajectories (see also Supplementary Figure 8-26 in Lagain et al., 2021). This makes me think that abrupt changes in excavation angles or spall fragments in rotation are uncommon processes on planetary bodies. The majority of homeless secondary crater rays or background secondaries with untrackable source might be due to (1) the erosion or the high dispersion of some portions of the rays or (2) the obliteration of their crater source, the former making hard the azimuth measurement of the rays. Moreover, in the hypothesis that the rotation of spall fragments had an influence in the direction of the rays, this should be observed more often on Mars than on airless bodies such as the Moon or Mercury. This is not the case.

A. Lagain et al., 2021. The Tharsis mantle source of depleted shergottites revealed by 90 million impact craters. *Nature Communications*, **12**, 6352. doi: 10.1038/s41467-021-26648-3.

- 2- L.81-83: “Therefore, the curved ray was formed by distal downrange ejecta that were excavated by the oblique impact that formed Hokusai¹⁹, with ballistic ranges over 4500 km from the crater center.”

I think this statement needs to be placed later in the manuscript. As currently written in the results section, the sum of evidences presented so far does not allow to prove this (the end of the results section might be a better place). Also, calling a figure showing Hokusai crater location on a global map and a close-up is definitely needed here.

- 3- L.205-208: “Rotational segmentation of spall fragments during flight, possibly caused by the residual strain energy and/or particle velocity differences in incipient spall plates²¹, could explain the diverged orientations of such rays (Fig. 3). This mechanism is also plausible to explain the origin of circular and spatially-isolated secondaries outside of radial rays of Hokusai²⁰.”

I do not see anything in the references cited here that support this hypothesis, in particular on an airless body. Could you please be more specific?

- 4- L.209-214: “Planetary impacts naturally involve materials with heterogeneous shock impedances, such as lithological and structural discontinuities of impactor and target materials. Therefore, abrupt changes of spallation angles should be an inherent characteristic of planetary impacts, explaining recent discoveries of self-secondaries on rims of simple craters on the Moon^{31,32} and predicting a common occurrence of background secondaries with untrackable provenances on planetary surfaces.”

My first comment also applies here.

Moreover, self-secondaries are interpreted as being “formed by material launched into near-vertical trajectories and having velocities such that their flight time is sufficiently long that the bulk of the clastic ejecta and impact melt are deposited before that material impacts the surface.” (Plescia and Robinson, 2019). The process the authors invoke in the present study to explain the curved ray of Hokusai crater is unrelated to self-secondaries on rims or ejecta blanket observed on other impact craters. Unless the authors clearly justify this, this sentence and any other references to self-secondary cratering should be removed in the revised version of the manuscript.

- 5- Fig.3 is of importance to illustrate the main claim of this study. However, I found it not that informative as it is. I think it needs to be annotated to guide the reader in its understanding. Detailed description of each panel would be useful too.

- 6- L.231-234: Azimuthally non-uniform ejection angles of spall fragments yield different flight times for ejecta that are landed at a same position (Supplementary Fig. 11). Therefore, spatial densities and crosscutting relationships of distal ejecta are not always reliable to compare relative ages of terranes on a same planetary body.”

Why? If their common source can be determined (which is most of the time the case), then crosscutting relationships of distal ejecta is therefore useful in determining relative ages of non-adjacent units. Also, I do not think that mentioning “spatial densities” of “distal ejecta” is appropriate here as this is never used to derive relative ages of terranes. See also my next comment.

- 7- Fig.4: Displaying the DEM on panel a in addition to the points color-coded based on the azimuth makes the figure very hard to read. Please annotate Divalia Fossae on panels a and b for clarity. The study case of Rheasilvia would deserve to modify panel b to better show the correspondence between landing position of simulated particles and circumferential chains of secondaries. It is not obvious to see as it is, especially for non-experts readers. I suggest to show close-ups of mapped circumferential chains along with simulated results.

Panel c is again hard to read. Radial ejecta are not obvious to see.

L:262-268: "An example of non-trackable background secondaries on Europa that could cause conflicted geological interpretations compared to the observed crosscutting relationships. The linear tectonic belt (green shade) is older than the smooth pool that is emplaced on older terrains (blue shade), as evident by the crosscutting relationship with nearby fractures, but the smooth pool features a larger spatial density of circular craters that do not resemble typical secondaries, although most small craters on Europa were interpreted to be secondaries formed by Pwyll³⁶." In ref.36 nor in the present study is shown a model age derivation of the two units by excluding clustered secondary craters. The green area seems also to be younger than the blue one (see figure below). Therefore, the above statement does not stand.

Minor comments

- Supp fig.1, add an arrow to point the hyperbolic ray
- L.55-57: "However, images returned by Mariner 10 were not adequate to conclude that the curved ray was indeed formed by a single impact, nor that the source crater was located at the hypothesized location¹⁶." Why is that ? Because of the unimaged portion of the surface at that time? Please be more precise.
- At first occurrence of Hokusai crater, please mention its coordinates and show it on a global map (supp fig.1 or add a panel to the fig.1)
- L.62:"However, the sharp change of orientation was not caused by the poised larger spin speed of Mercury, but due to azimuthally non-uniform ejection angles during impact spallation." I would briefly define "spallation" for readers not expert in cratering processes.

- Supp fig.2, I do not understand the presence of the white circles at the antipodal points of the three craters investigated on each panels.
- General comment: increase the resolution of all figures, especially for sup. Fig.7 and 12
- Fig.4.d does not show any grid or coordinates information.

Data availability:

I appreciate the authors made available the updated mapping of Hokusai's rays, model output of ejecta landing, and some of the images used in this work.

However, I have not been able to open those images. Could you please update the figshare repo by changing their format into .geotiff or similar instead of .7z.001 .7z.002 and so on? Also, the authors mentioned that "The code used to model ballistic trajectories of impact ejecta on airless planetary bodies is available from the corresponding author upon request." I would greatly appreciate to have a look to the code to check the ejecta trajectory modelling.

My name can be revealed to the authors.

Anthony Lagain

“Untrackable distal ejecta on planetary surfaces”

by Rui Xu and Zhiyong Xiao, Fanglu Luo, Yichen Wang, Jun Cui

[Paper # NCOMMS-22-38439]

Response to Reviewers

The reviewers' comments are in black Times New Roman font. My responses are in blue Arial.

REVIEWER COMMENTS

Reviewer #1 (Remarks to the Author):

This manuscript discusses about the origin of non-trackable distal ejecta on Mercury, Vesta and Europa. The authors demonstrated that the variation of the ejection angle during impact ejection can explain the “curved ray”, which is one of the outstanding problems on the geology on Mercury. The authors also ruled out two attractive hypotheses on the origin of the curved ray, which are 1) fast-spinning Mercury at the past and 2) unknown source crater by the detailed surface observation on Mercury and trajectory modeling. The authors proposed that relatively high-ejection angle can be explained by the process called impact spallation. Finally, the authors concluded that distal ejecta launched from single crater would produce complex relationships which tend to lead wrong stratigraphic interpretations.

The authors appreciate the comments provided by the reviewer, which are very helpful and constructive. All the comments and suggestions are integrated point by point in this revision.

The manuscript is not well written and not well organized. The presentation quality of the manuscript is poor especially for inter-discipline journals, such as Nature Communications (See major comment 1 and 2). The reviewer, however, agrees with the authors that the finding is important to accurately interpret the chronology of the planetary surfaces. In addition, the reviewer does not agree with the author's statement that the relatively high ejection angles come from the local topography around impact site (See major comment 3). Consequently, the reviewer supports the publication of the manuscript, but after a major revision. The reviewer also made comments on the word file.

The authors appreciate the reviewer's positive evaluation on the scientific value of this work. Thank you for all the comments and suggestions.

The authors agree with this reviewer that the organization and presentation of this manuscript need systematically revision. The authors have followed the suggestions and completely rewritten the manuscript. The resolution and design of images used in the initial submission were not good enough to show

details. The authors now systematically revised the presentations, which should be clearly visible when printed on A4 (or Letter Size) papers.

With a clearer structure and presentation of this revision, the authors wish to demonstrate that the modelled abrupt changes of ejection angles occurred during impact spallation, and such changes are not systematic at different azimuths (e.g., downrange versus uprange of oblique impacts). This work suggests that factors that may affect the local propagation paths of shock and rarefaction waves are the reason, such as topographic undulations that have appropriate geometries.

There are five major comments as follows. Minor comments are made in the attached files.

Each of the comments is considered and adopted in revising the manuscript.

Major comment 1

The organization of the manuscript is not good. The current manuscript has been hard to understand, at least for the reviewer, what the authors would like to state. The reviewer had to repetitively read the manuscript to understand the content. The reviewer strongly recommends to largely revise the organization of the manuscript to enhance the values of the author's findings.

The source of the confusions (at least for the reviewer) is the lack of a clear separation between what can be explained by existing knowledge and what is revealed in this study. The former is the radial ray system on Mercury, Vesta and Europa, and the latter is the origin of the curved ray on Mercury, the origin of troughs on Vesta, and the origin of groove-like structure on Europa. In addition, the authors mentioned about the observational fact and the prediction from impact physics without separation.

The reviewer proposes a better organization, for example, as follows.

Introduction about distal ejecta

- Radial ray system
- There are non-trackable ejecta deposits on Mercury, Vesta, and Europa

High-spin rate has been proposed in previous studies

Model results

- Radial crater rays can be explained by the conventional knowledge about impact cratering

(~45 degree of ejection angle)

- The test of the high-spin rate hypothesis.

It can explain the non-radial ejecta pattern on Europa, but cannot explain the curved

ray on Mercury.

•High-ejection angle can explain the deposition of the curved ray.

Of course, the authors do not necessarily follow above completely, but the current manuscript is difficult to read because it is just a list of case studies for each object. A drastic change in the structure of the paper may dramatically improve the readability of the manuscript.

The authors appreciate this comment very much, which is fair and constructive. Thank you.

The organization of this manuscript is now reorganized to better show the logic. Several background points that stimulate this study, which were not clearly presented in the initial submission, are briefly introduced here for clarify

(1) the curved impact ray on Mercury was noticed in images obtained by the Mariner 10 spacecraft, and it was interpreted to be caused by a larger spin rate of Mercury in the near past;

(2) Mariner 10 data covered ~44% of Mercury with an average pixel scale of ~1000 m, and the potential source crater was postulated to be located at the unimaged hemisphere of Mariner 10;

(3) impact craters with obvious rays are widespread on both Mercury and the other planetary bodies, and the Mariner 10 data were not adequate in terms of both pixel scales and spectral resolutions to determine that the curved ray was indeed formed by a same impact crater;

(4) earlier studies of impact rays and other distal ejecta deposits on 4Vesta and Europa did not particularly focus on non-radial ejecta; in literatures, there was no debate neither about possible age differences between smooth plains on Europa, or whether or not the troughs on 4Vesta were formed by secondary impacts; however, the origin of the troughs on 4Vesta and formation time of smooth plains on Europa are important questions in planetary science, and the discovery of this work could provide additional insights into these questions;

(4') based on our observations and modelling, this work predicts common occurrences of non-radial distal ejecta on various planetary bodies, especially on fast rotating bodies; this work shows that at least some troughs parallel to the Divalia Fossae on 4Vesta are consistent with being chains of secondaries formed by the Rheasilvia basin, so these troughs are not necessarily tectonic features;

(4'') this study predicts overlapped landing of ejecta that had different ejection conditions from a same source crater, so that spatial densities of impact craters (which are not typical secondaries) are not always safe to represent relative/absolute model ages of geological units; this work now takes an example on Mars for the demonstration: non-radial distributions of "rays" of

distant secondaries commonly exist around young craters on Mars, and some “rays” from a same crater exhibit overlapping relationships (Lagain et al., 2021); a single lava flow passed through the intersection region of two such rays that were formed by the Corinto crater, and the crater densities exhibit large variations on the same lava flow.

With the above background and aspiration, the authors summarize the revised structure of this manuscript:

(1) Introduction: conventional knowledge about radial distribution of impact ejecta; importance of radial ejecta in constructing planetary stratigraphy and untangling provenances of planetary surface materials; original interpretation of the curved ray on Mercury; main shortages in earlier works.

(2) Results 1: new observations using MESSENGER WAC multi-band images to prove that the curved ray was formed by the Hokusai crater; new observations using high-resolution MESSENGER MDIS NAC images to prove that secondaries in the western branch of the curved ray were ejected from a same azimuth, and those in the eastern branch of the curved ray were also ejected from a same azimuth (but not necessarily the same with the western branch); comparison with the morphology of secondaries in the other non-radial rays of Hokusai and those on the other planetary bodies, showing that the curved ray in question was not formed by segments of rays that each has a radial distribution with respect to Hokusai.

(3) Results 2: introduce model construction and validation; model the global distribution of Hokusai’s rays, showing that most visible rays are consistent with the conventional knowledge of impact ejection ($45^{\circ}\pm 15^{\circ}$) and the curved ray is an outlier.

(4) Results 3: for the global ray system of Hokusai, model landing positions of impact ejecta based on different assumed ejection conditions, showing that the curved ray cannot be caused by a faster rotation of Mercury; for the curved ray in question, model landing positions of impact ejecta based on different assumed ejection conditions, showing that the western and eastern branches cannot be formed by ejecta with a same ejection azimuth, yielding the best-fit ejection conditions.

(5) Discussion: prove that curved ray was formed by ejecta with abnormal and sudden change of ejection angles; from the perspective of shock mechanics reported in earlier studies, discuss possible causes for the required steep ejection angles.

(6) Predictions: use three outstanding cases to demonstrate the geological implications of this discovery, (a) once non-radial ejecta are degraded with time, many distal ejecta would have unobvious provenances on planetary surfaces; the origin of troughs parallel to the Divalia Fossae on 4Vesta, which are generally regarded as tectonic features; observations and modelling in this

work provide an alternatively interpretation that they can be chains of secondaries; the second example is the provenance of exotic materials in the Chang'E-5 regolith, which is currently a debate; (b) this work shows that distal ejecta formed by a same crater may land at the same location but at different times and from different azimuths, especially on fast rotating planetary bodies such as Mars; use a case on Mars to demonstrate that spatial densities of impact craters may be not always safe to represent relative or absolute model ages.

Reference:

Lagain, A. et al. The Tharsis mantle source of depleted shergottites revealed by 90 million impact craters. *Nat. Commun.* **12(1)**, 1–9 (2021).

Major comment 2

It has been hard to interpret the current figures at least for the reviewer. Although in some respects it is inevitable that the figures have many information, it seems to be necessary to modify it. The reviewer recommends to revise the figures to be that can be read at single glance, or to add supplementary figures. The reviewer made specific comments in the word file. Please examine the separated file in detail.

The authors appreciate this comment, which urges us to improve the quality of presentation. The color schemes of the figures are now revised to better show the content, and full-resolution images are now used in this revision, so that detailed information is straightforward and obvious as seen on printed papers.

Major comment 3

The authors stated “Recent numerical simulations revealed pronounced effect of stochastic target topography undulations on spallation angles, and interaction between shock waves and local topography causes locally steepened spallation angles. It is educible that other forms of structural discontinuities with appropriate geometries, such as lithological boundaries, wrinkle ridges, and/or unflooded crater rims in the northern volcanic plains of Mercury where Hokusai was formed, could change local propagation paths of shock and rarefaction waves, causing abrupt changes of spallation angles (Fig. 3).”. The reviewer believes that the relatively-high ejection angles can be explained solely from an impact ejection during oblique impacts. Okamoto et al. (2020), *JGR-Planets*, 125, e2019JE005943 conducted both hypervelocity oblique-impact experiments and shock physics modeling and demonstrated that the ejection angle depends on azimuth angle (See Figs. 11 and 12 of Okamoto et al., 2020).

Combined with the author’s model, the west–east branch of curved ray on Mercury might provide the information of the impact direction of the impactor, which produced Hokusai crater. The reviewer thinks that the above possibility is worthwhile to add the discussion section to enhance the value of the manuscript.

The authors understand this concern.

Okamoto et al. (2020) successfully combined physical and numerical simulations to study impact jetting and early spallation during oblique impacts. Significant insights were provided about the evolution of ejection velocities and angles from both target and projectile materials (Figures 11 and 12; Okamoto et al., 2020). The results showed that (1) spalls with extremely large ejection angles ($\theta \cong 0^\circ$ with respect to the surface normal) and velocities up to $\sim 50\%$ of the impact velocity exist, and most of such spalls are concentrated towards the uprange of the impactor trajectory; (2) downrange ejecta that include jets and spalls feature much larger velocities and shallower ejection angles ($\theta < 45^\circ$ from the surface tangent).

In this work, the authors try to demonstrate that the curved ray on Mercury was formed due to suddenly-steepened ejection angles during impact spallation, and the model-predicted change of ejection angles only occurred at certain azimuths and ejection velocities. While the Hokusai crater was formed by an oblique impact, the majority of its global ray system was radial to the crater and their ejection conditions are consistent with model predictions based on normal ejection conditions (i.e., $\theta \cong 45^\circ$). This study reports no systematic variations in conditions of ejection for the distal rays, and the oblique impact angle of Hokusai was not an obvious reason. This discussion is now added in the revision for explanation (lines 250 – 260).

On the other hand, the authors noticed that Okamoto et al. (2020) demonstrated that spalls launched towards the uprange have relatively small ejection velocities ($v_e/v_i < 0.5$) and near vertical ejection angles (Figure 12 of Okamoto et al., 2020). This observation, if applicable to the Hokusai impact, could potentially explain the observation that self-secondaries on continuous ejecta deposits of Hokusai exhibit a slightly larger spatial density at the uprange. The formation of these secondaries also require near vertical ejection angles of spall fragments. By citing the work of Okamoto et al. (2020), this possibility is now added in the discussion (lines 287 – 295).

Reference:

Okamoto, T., Kurosawa, K., Genda, H. & Matsui, T. Impact ejecta near the impact point observed using ultrahigh-speed imaging and SPH simulations and a comparison of the two methods. *J. Geophys. Res. Planets* **125**, e2019JE005943 (2020).

Major comment 4

Kadono et al. (2020), which has been referred in the manuscript mentioned that the crater rays formed as a consequence of inelastic collision between ejecta particles in an ejecta curtain, and that a mesh pattern between adjoining rays occurs during the growth of ejecta (Fig.2 in Kadono et al. 2020). Nakazawa et al. (2021), *The Planetary Science Journal*, 2:237, further explored the formation mechanism with both impact experiments and N-body simulations. They demonstrated that the mesh pattern

formation can be explained by the clustering process of ejecta particles and pattern stretching during ballistic flight would form crater-ray structure. The reviewer is wondering that the non-radial crater rays investigated by the authors might be relics of the mesh pattern. The authors should address such another possibility because the formation of crater rays proceeds via mesh pattern formation according to the previous studies (Kadono et al., and Nakazawa et al.).

The authors agree with the reviewer that during ballistic flight, inelastic collisions and the related pattern stretching among ejecta particles could form non-radial rays, which were nicely demonstrated by Kadono et al. (2015, 2020) and Nakazawa et al. (2021).

The authors also note that non-radial rays are widespread on planetary bodies, and many of them can be explained by earlier-proposed mechanisms such as oblique impacts and pattern stretching of ejecta during flight. Besides lateral extensions of impact rays, the azimuth of landing of ejecta that formed such non-radial rays are frequently ignored. However, this information is critical to determine the flight trajectory of ejecta.

One of the innovations of this work is combining high-resolution morphological study of secondaries and numerical simulations for landing patterns of ejecta to determine the azimuth of landing. This strategy is reasonable thanks to earlier physical simulations for the formation of secondaries (e.g., Schultz and Gault, 1985), as typical secondaries exhibit higher elevations of rims at the uprange, subdued rims at downrange, and preferential deposition of ejecta towards the downrange. Based on such observations, modelling performed in this work show that while many ejecta of Hokusai can land on the curved ray, those landed from a same azimuth along the ray required specific ejection conditions.

Reference:

Kadono, T. et al. Crater-ray formation by impact-induced ejecta particles. *Icarus* **250**, 215–221 (2015).

Kadono, T. et al. Crater-ray formation through mutual collisions of hypervelocity-impact induced ejecta particles. *Icarus* **339**, 113590 (2020).

Nakazawa, K., Okuzumi, S., Kurosawa, K. & Hasegawa, S. Modeling Early Clustering of Impact-induced Ejecta Particles Based on Laboratory and Numerical Experiments. *Planet. Sci. J.* **2(6)**, 237 (2021).

Schultz, P. & Gault, D. E. Clustered impacts: Experiments and implications. *J. Geophys. Res.* **90(B5)**, 3701–3732 (1985).

Major comment 5

The undefined term “background secondaries” was used repetitively throughout the manuscript.

In addition, it is not clear throughout the manuscript what this term refers to. The authors should explicitly define the term in abstract if this term is so important.

The authors agree with this reviewer that discussions about background secondaries (a subpopulation of secondaries that are circular in shapes and related isolated in spatial distributions, similar to same-sized primaries) should not be over-emphasized based on the results of this study. This revision now substantially reduces the related discussions.

Thank you for the comments and suggestions.

Reviewer #2 Dr. Anthony Lagain (Remarks to the Author):

In this study, the authors apply ejecta modelling to investigate deflection from radial trajectory of secondary crater rays observed on Mercury. The authors evaluated the influence of debris excavation location with respect of crater centroid, ejection angle and speed as well as the rotational speed of the planet. They show that (1) a major curved ray observed on the surface of Mercury cannot be formed by any other impact craters than Hokusai, and (2) that abrupt changes in ejection angle of spall debris can explain the morphology of curved rays, deviating from radial trajectory. Thus, this study rules out the effect of the Coriolis effect as the main process forming such secondary rays, even on fast rotating bodies.

Overall, I found this paper well written and presented (although this could be improved in some aspects). Modelling and parameters investigated are appropriate, and the main conclusions are supported by the data. Topic is original and outputs have strong implications in the understanding of cratering process, planetary surface evolution and dating, and material displacement from impact cratering. However, I have a few major points I would like to see addressed by the authors before the study gets published.

The authors appreciate the overall positive evaluation by Dr. Lagain.

The comments and suggestions are very helpful and provoking, and the manuscript is now revised after integrating each of the suggestions. Thank you.

Major comments:

- 1- L.63-66: “Analyses of cratering mechanics predict that such homeless distal ejecta should be common on planetary surfaces, supporting a widespread occurrence of background secondaries and hampering reliable stratigraphic interpretations based on crosscutting relationship of distal ejecta.

If homeless distal ejecta should be common on planetary surfaces, why we do not see such hyperbolic rays associated with the largest and youngest craters formed on Mars (Lagain et al., 2021, <https://hive.curtin.edu.au/research/CDA-94M-release/>)? At least 19 craters associated with secondary crater rays are presented in this work, with rays

extending hundreds to thousands kilometres away from their crater of origin. None of them present significant deflection from radial trajectories (see also Supplementary Figure 8-26 in Lagain et al., 2021). This makes me think that abrupt changes in excavation angles or spall fragments in rotation are uncommon processes on planetary bodies. The majority of homeless secondary crater rays or background secondaries with untrackable source might be due to (1) the erosion or the high dispersion of some portions of the rays or (2) the obliteration of their crater source, the former making hard the azimuth measurement of the rays. Moreover, in the hypothesis that the rotation of spall fragments had an influence in the direction of the rays, this should be observed more often on Mars than on airless bodies such as the Moon or Mercury. This is not the case.

A. Lagain et al., 2021. The Tharsis mantle source of depleted shergottites revealed by 90 million impact craters. *Nature Communications*, 12, 6352. doi: 10.1038/s41467-021-26648-3.

The authors appreciate the reviewer's mention on this important reference. Lagain et al. (2021, 2022) have done a remarkable work to identify possible source craters of various martian meteorites. Tracking parent craters using the global catalog of smaller impact craters is a fabulous exercise.

First of all, the authors agree with the reviewer that topography degradation of both impact rays and parent craters could form untrackable secondaries, especially on Mars. This basic fact is emphasized in the manuscript, from the perspective of abnormal ejection conditions that permit the formation of non-radial ejecta.

Second of all, the authors agree with the reviewer that the majority of impact rays and other forms of distal ejecta on planetary bodies follow radial distributions around their source craters. The point of this work is that many non-radial rays (e.g., 6% of the global ray system of the Hokusai crater) were formed by ejecta with abnormal ejection conditions.

Third of all, the authors agree with the reviewer that the results of this study are not adequate to absolutely support that rotational fragmentation of spalls is an observed mechanism to form non-radial rays. This discussion is now omitted.

The authors noticed that most young craters with thermophysical rays on Mars also formed a minor portion of non-radial rays (e.g., Fig. R1), and some of them were mentioned, but not particularly investigated in earlier studies (e.g., the caption of Figure 5 in Tornabene et al., 2006; Figure 2 of Schultz et al., 2009). Curved ejecta that have non-radial orientations are also observable in chains of near-field secondaries on Mars (Fig. R2). However, these are not the same type of ejecta as those investigated in this work (i.e., near-field versus distal ejecta), and the morphology of the secondaries shown in Fig. R2b suggests that each of the secondary craters had a radial azimuth of landing. In this revision, a case of non-radial rays formed by an unnamed crater ($D=18.5$

km) is added (Fig. 2 c, f), and the purpose is to demonstrate that similar with cases on the Moon and Mercury, such non-radial rays were formed by ejecta that can be tracked radially back to the source crater. The morphology of secondaries and their associated ejecta are used as key evidence for this argument, which was based on physical simulations for the formation of secondaries (Schultz and Gault, 1985). Such non-radial rays were likely formed due to inelastic collisions of ejecta during flight (Kadono et al., 2015; Kadono et al., 2020; Nakazawa et al., 2021) or the oblique impact angle that formed the source crater (Schultz et al., 2009), and the ejection conditions may be different with those formed the curved ray on Mercury.

Fig. R1 | Non-radial rays are commonly observed around young martian craters that have thermophysical rays, and most rays follow radial distributions. The four cases used here are the Dilly (rim-to-rim diameter $D = 2.0$ km), Zumba ($D = 3.3$ km), Tomini ($D = 7.4$ km), and Gratteri ($D = 6.9$ km). The base images are from the THEMIS global night mosaics.

Fig. R2 | A curved chain of secondaries is visible to the southeast of an unnamed crater ($D=18.5$ km) on Mars. The raised uprange rims, subdued downrange rims, and preferential deposition of ejecta formed by the secondaries towards the downrange suggest that the ejecta that formed the secondaries had radial azimuths of landing with respect to the source crater. This crater has an ID of 09-000015 in Lagain et al. (2021). The base image is from the CTX mosaic MLab09000015 (Murray Lab, CalTec).

Following the link kindly provided by the reviewer (<https://hive.curtin.edu.au/research/CDA-94M-release/>) and the supplementary figures (Figs. 8–26) reported by Lagain et al. (2021), the authors also noticed that in the recognized “rays” of secondaries, non-radial “ray segments” are widespread, albeit a minor portion among the detected “rays” (most are radial). Several such cases are annotated below in Fig. R3. The authors made georeferencing calibration for the reported figures, since the original datasets were not available online. Correlated observations in CTX mosaics reveal that most such non-radial “rays” of secondaries are quite circular and relatively isolated in the spatial distribution, which are unlike the morphology of typical secondaries (Fig. R3). This observation echoes earlier discovery that secondaries on Mars are more circular in shapes than lunar craters (Schultz and Singer, 1980). Azimuths of landing of ejecta that formed these “non-radial rays” of secondaries are usually unobvious based on the morphology of these secondaries, so their ejection conditions are not straightforward. Nevertheless, curved rays and distant secondaries that have non-radial distributions are not uncommon on Mars.

Fig. R3 | Non-radial distributions of “rays” of secondaries reported by Lagain et al. (2021).

For a further discussion with Dr. Lagain, which is not directly related with this work, the authors deeply believe that the work of Lagain et al. (2021) has a great potential to advance the current understanding of possible contamination of background secondaries to crater chronology (this is currently a major obstacle in planetary science). This dataset is simply fabulous.

Reference:

Lagain, A. et al. The Tharsis mantle source of depleted shergottites revealed by 90 million impact craters. *Nat. Commun.* **12**(1), 1–9 (2021).

Lagain, A. et al. Early crustal processes revealed by the ejection site of the oldest martian meteorite. *Nat. Commun.* **13**(1), 1–8 (2022).

Tornabene, L. L. et al. Identification of large (2–10 km) rayed craters on Mars in THEMIS thermal infrared images: Implications for possible Martian meteorite source regions. *J. Geophys. Res. Planets* **111**(E10) (2006).

Schultz, P. H., Anderson, J. L. B. & Hermalyn, B. Origin and significance of uprange ray patterns. *LPSC 2496* (2009).

Schultz, P. & Gault, D. E. Clustered impacts: Experiments and implications. *J. Geophys. Res.* **90(B5)**, 3701–3732 (1985).

Kadono, T. et al. Crater-ray formation by impact-induced ejecta particles. *Icarus* **250**, 215–221 (2015).

Kadono, T. et al. Crater-ray formation through mutual collisions of hypervelocity-impact induced ejecta particles. *Icarus* **339**, 113590 (2020).

Nakazawa, K., Okuzumi, S., Kurosawa, K. & Hasegawa, S. Modeling Early Clustering of Impact-induced Ejecta Particles Based on Laboratory and Numerical Experiments. *Planet. Sci. J.* **2(6)**, 237 (2021).

Schultz, P. & Singer, J. A comparison of secondary craters on the Moon, Mercury, and Mars. *LPSC 2243–2259* (1980).

2- L.81-83: “Therefore, the curved ray was formed by distal downrange ejecta that were excavated by the oblique impact that formed Hokusai¹⁹, with ballistic ranges over 4500 km from the crater center.”

I think this statement needs to be placed later in the manuscript. As currently written in the results section, the sum of evidences presented so far does not allow to prove this (the end of the results section might be a better place). Also, calling a figure showing Hokusai crater location on a global map and a close-up is definitely needed here.

The authors agree with this comment.

The map of the global ray system of Hokusai is now reported in Fig. 1a, and the summary that the curved ray was formed by downrange ejecta of Hokusai is now placed afterward.

3- L.205-208: “Rotational segmentation of spall fragments during flight, possibly caused by the residual strain energy and/or particle velocity differences in incipient spall plates²¹, could explain the diverged orientations of such rays (Fig. 3). This mechanism is also plausible to explain the origin of circular and spatially-isolated secondaries outside of radial rays of Hokusai²⁰.”

I do not see anything in the references cited here that support this hypothesis, in particular on an airless body. Could you please be more specific?

The authors agree with the reviewer that the results reported here are not adequate to absolutely support that rotational fragmentation of spalls is an additional mechanism to form non-radial rays.

The calculation of Melosh (1984) showed that a single spall fragment launched from the surface had different ejection velocities at different parts, setting spinning about various axes. Rotational fragmentation may occur if the centrifugal force exceeds the tensile strength of the spall fragment. Xiao (2016) followed this theory to explain the observation that at larger distances

perpendicularly from an impact ray, secondaries show a decreasing trending in the spatial resolution.

The original discussion about rotational fragmentation is now deleted.

Reference:

Melosh, H. J. Impact ejection, spallation, and the origin of meteorites. *Icarus* **59(2)**, 234–260 (1984).

Xiao, Z. Size-frequency distribution of different secondary crater populations: 1. Equilibrium caused by secondary impacts. *J. Geophys. Res. Planets* **121**, 2404–2425 (2016).

4- L.209-214: “Planetary impacts naturally involve materials with heterogeneous shock impedances, such as lithological and structural discontinuities of impactor and target materials. Therefore, abrupt changes of spallation angles should be an inherent characteristic of planetary impacts, explaining recent discoveries of self-secondaries on rims of simple craters on the Moon^{31,32} and predicting a common occurrence of background secondaries with untrackable provenances on planetary surfaces.”

My first comment also applies here.

Moreover, self-secondaries are interpreted as being “formed by material launched into near-vertical trajectories and having velocities such that their flight time is sufficiently long that the bulk of the clastic ejecta and impact melt are deposited before that material impacts the surface.” (Plescia and Robinson, 2019). The process the authors invoke in the present study to explain the curved ray of Hokusai crater is unrelated to self-secondaries on rims or ejecta blanket observed on other impact craters. Unless the authors clearly justify this, this sentence and any other references to self-secondary cratering should be removed in the revised version of the manuscript.

The original manuscript was designed as a short letter, and the descriptions and discussions were compressed in length, which have caused confusions. Self-secondaries were formed by near-vertically ejected spall fragments, and for the Hokusai case, observations and calculations by Xiao et al. (2016) suggested the self-secondaries were launched with velocities $>\sim 1$ km/s and angles less than 10 degrees from the surface normal. Modelling in this work shows that spalls with comparable ejection velocities mostly occur as normal radial rays (and their interior secondaries) around Hokusai. Therefore, the existence of self-secondaries on continuous ejecta deposits of Hokusai supports that abrupt changes of ejection angles occurred at different ejection velocities during impact spallation.

Reference:

Xiao, Z., Prieur, N. C. & Werner, S. C. The self-secondary crater population of the Hokusai crater on Mercury. *Geophys. Res. Lett.* **43**, 7424–7432 (2016).

5- Fig.3 is of importance to illustrate the main claim of this study. However, I found it

not that informative as it is. I think it needs to be annotated to guide the reader in its understanding. Detailed description of each panel would be useful too.

The manuscript is now expanded as a full paper following the suggestions and comments provided by both of the reviewers. With a clearer explanation, the formation mechanism of non-radial ejecta that is manifested in the schematic diagram should be more illustrative. The figure caption is also reorganized accordingly to better show the meaning of the diagram.

6- L.231-234: Azimuthally non-uniform ejection angles of spall fragments yield different flight times for ejecta that are landed at a same position (Supplementary Fig. 11). Therefore, spatial densities and crosscutting relationships of distal ejecta are not always reliable to compare relative ages of terranes on a same planetary body.”

Why? If their common source can be determined (which is most of the time the case), then crosscutting relationships of distal ejecta is therefore useful in determining relative ages of non-adjacent units. Also, I do not think that mentioning “spatial densities” of “distal ejecta” is appropriate here as this is never used to derive relative ages of terranes. See also my next comment.

The authors agree with the reviewer that the original manuscript did not clearly pass the message that distal ejecta formed by a same crater could be land at the same location but at different times and from different azimuths.

In this revision, the discussion is expanded accordingly. The theory that leading this prediction (mainly due to a combination of different v , θ and ϕ) and observational evidence based on a case study on Mars (mainly from the observation by Lagain et al. 2021 and correlated observations on a same lava flow) are introduced to backup this argument.

Reference:

Lagain, A. et al. The Tharsis mantle source of depleted shergottites revealed by 90 million impact craters. *Nat. Commun.* **12(1)**, 1–9 (2021).

7- Fig.4: Displaying the DEM on panel a in addition to the points color-coded based on the azimuth makes the figure very hard to read. Please annotate Divalia Fossae on panels a and b for clarity. The study case of Rheasilvia would deserve to modify panel b to better show the correspondence between landing position of simulated particles and circumferential chains of secondaries. It is not obvious to see as it is, especially for non-experts readers. I suggest to show close-ups of mapped circumferential chains along with simulated results.

Panel c is again hard to read. Radial ejecta are not obvious to see.

The authors have followed this suggestion and revised this figure to better show the details. Also, the discussions (possible origin of secondary impacts for the troughs on 4Vesta) are now reorganized, and the case on 4Vesta is used as an implication to show the geological significance of non-radial ejecta

on planetary bodies.

L:262-268: “An example of non-trackable background secondaries on Europa that could cause conflicted geological interpretations compared to the observed crosscutting relationships. The linear tectonic belt (green shade) is older than the smooth pool that is emplaced on older terrains (blue shade), as evident by the crosscutting relationship with nearby fractures, but the smooth pool features a larger spatial density of circular craters that do not resemble typical secondaries, although most small craters on Europa were interpreted to be secondaries formed by Pwyll³⁶.” In ref.36 nor in the present study is shown a model age derivation of the two units by excluding clustered secondary craters. The green area seems also to be younger than the blue one (see figure below). Therefore, the above statement does not stand.

The reviewer points out a caveat in the initial submission about the logic of demonstration for the case on Europa: in the initial submission, the claimed larger spatial density of impact craters on the smooth plain was not really established; if the claimed larger density was true, it can be alternatively caused by other factors, such as a fast destruction rate of craters in the grooved terrane (which is believed to be tectonically active). The authors now removed this case and the related discussion in this revision.

In this work, the modelling shows that ejecta formed by a same crater that have different combinations of v , θ and ϕ can land at a same location but at different times and from different azimuths. The orientational inflection branch of the curved ray on Mercury should be a good example to demonstrate this prediction (Fig. 4c), but available high-resolution images are not good enough to investigate the spatial density or overlapping relationship of secondaries in the ray. The chance of discovering larger densities of secondaries that are caused by overlapped landing of distal ejecta is larger on fast rotating planetary bodies, such as Mars and Europa. Europa should be an excellent target for this demonstration, because most small craters less than 1 km in diameters are believed to be formed by a single crater Pwyll (Bierhaus et al., 2005). However, as pointed out by the reviewer, the case reported in the initial submission was not convincing. Although Europa has an extremely young ice crust due to active tectonic deformation and resurfacing, surface topography is rather rough, which is almost full of groove bands and mosaic terrains. Moreover, only a handful frames of high-resolution images (better than 50 m/pixel) are available for Europa, so suitable cases for a successful demonstration is limited. Therefore, this revision gives up using cases on Europa but only mentions this possibility in the supporting information.

The nice contribution by Lagain et al. (2021) motivated the authors to use a case on Mars for the demonstration. Non-radial distributions of “rays” of distant secondaries commonly exist around young craters on Mars, and such “rays” from a same crater frequently exhibit overlapping relationships, e.g., the case reported in the Supplementary Fig. 14 of Lagain et al. (2021). This figure is

now pasted, and the curved “ray” and a more-or-less radial “ray” of secondaries are denoted by red and green arrows, respectively (Fig. R4). A single lava flow passed through the intersection region of two distant rays of secondaries formed by the Corinto crater (Fig. 10a, b). Most visible craters in the intersection region are circular in shapes and relatively isolated in the spatial distribution (Fig. 10c), which are not different from same-sized primaries. It has been noticed many secondaries on Mars feature such morphological and distributional characteristics (Schultz and Singer, 1980). Standard technical procedure of crater statistics would not exclude such craters when determining model ages for planetary surfaces (Michael and Neukum, 2010). This study selects two regions on the lava flow, one in the intersection region of the two rays of secondaries, and another one in the radial ray of secondaries. Dramatically different crater densities are observed (Fig. 10d), which are consistent with the predicted landing patterns of impact ejecta that have different combinations of v , θ and ϕ at the same location (Supplementary Fig. 7).

Fig. R4 | Non-radial “rays” of secondaries reported by Lagain et al. (2014). This case was reported in the Supplementary Fig. 14 of Lagain et al. (2021), and the red and green arrows denote a curved “ray” and a “radial” ray, respectively.

Reference:

Bierhaus, E. B., Chapman, C. R. & Merline, W. J. Secondary craters on Europa and implications for cratered surfaces. *Nature* **437(7062)**, 1125–1127 (2005).

Lagain, A. et al. The Tharsis mantle source of depleted shergottites revealed by 90 million impact craters. *Nat. Commun.* **12(1)**, 1–9 (2021).

Schultz, P. & Singer, J. A comparison of secondary craters on the Moon, Mercury, and Mars. *LPSC* 2243–2259 (1980).

Michael, G. G. & Neukum, G. Planetary surface dating from crater size–frequency distribution measurements: Partial resurfacing events and statistical age uncertainty. *Earth Planet. Sci. Lett.* **294**, 223–229 (2010).

Minor comments

- Supp fig.1, add an arrow to point the hyperbolic ray

Added.

- L.55-57: “However, images returned by Mariner 10 were not adequate to conclude that the curved ray was indeed formed by a single impact, nor that the source crater was located at the hypothesized location¹⁶.”

Why is that? Because of the unimaged portion of the surface at that time? Please be more precise.

The reason is now elaborated in this revision (lines 49–54).

- At first occurrence of Hokusai crater, please mention its coordinates and show it on a global map (supp fig.1 or add a panel to the fig.1)

The manuscript is now revised following this suggestion, and the location and coordinates of Hokusai is added in Fig. 1

- L.62: “However, the sharp change of orientation was not caused by the poised larger spin speed of Mercury, but due to azimuthally non-uniform ejection angles during impact spallation.”

I would briefly define “spallation” for readers not expert in cratering processes.

The introduction of impact spallation is now moved to the discussion section.

- Supp fig.2, I do not understand the presence of the white circles at the antipodal points of the three craters investigated on each panels.

This figure is no longer needed. These white “circles” were landing positions of ejecta that have very high velocities, which flew all around Mercury before landing.

- General comment: increase the resolution of all figures, especially for sup. Fig.7 and 12

Full resolution images are now used in this revision, and long images are now

separated to shorter ones.

- Fig.4.d does not show any grid or coordinates information.

This case is no longer used in this revision.

Data availability:

I appreciate the authors made available the updated mapping of Hokusai's rays, model output of ejecta landing, and some of the images used in this work.

However, I have not been able to open those images. Could you please update the figshare repo by changing their format into .geotiff or similar instead of .7z.001 .7z.002 and so on? Also, the authors mentioned that "The code used to model ballistic trajectories of impact ejecta on airless planetary bodies is available from the corresponding author upon request." I would greatly appreciate to have a look to the code to check the ejecta trajectory modelling.

All the data produced in this paper are now provided for open access, which are available at <https://doi.org/10.6084/m9.figshare.20722891.v4>.

The ballistic trajectory code is now opened for public access as well: <https://doi.org/10.6084/m9.figshare.20722891.v4>.

Thank you again for all the comments and suggestions.

Responses completed.

REVIEWERS' COMMENTS

Reviewer #1 (Remarks to the Author):

Review for NCOMMS-22-38439A

Untrackable distal ejecta on planetary surfaces

Written by Rui Xu and Zhiyong Xiao, Fanglu Luo, Yichen Wang

The reviewer largely appreciates the authors. The reviewer confirmed that the authors adequately addressed all my concerns. The reviewer believes that the revised manuscript clearly demonstrates the significance of the non-trackable distal ejecta on the understanding the regional and global stratigraphy on planetary bodies. The reviewer notes only some very minor points for the final revision before submission (which I do not need to see again).

L253–254:

“Spall fragments excavated by oblique impacts exhibit azimuth asymmetry in the ejection velocities and angles³⁵.”

The reference #33 (oblique impacts) should be cited in this context, not #35 (vertical impacts only).

L263–264:

Sabuwala et al., 2018(#37) also conducted a series of laboratory impact experiments. Please cite the previous study more accurately.

L270:

What is “ample condition”? Is this typo?

Figure 9b

The authors should indicate the locations of the faint impact rays with several arrows or something like that.

Supplementary Figs. 6 and 7:

The authors need to explain the meaning of the color in the figure. I recommend that a color bar would be added beside the figure.

Reviewer #2 (Remarks to the Author):

The authors have addressed each one of my concerns in detail. This revised manuscript is better presented, clearer, the implications well-justified, and study cases convincing. In my opinion the manuscript is ready for publication.

Anthony Lagain

Nature Communications
“Untrackable distal ejecta on planetary surfaces”
by Rui Xu and Zhiyong Xiao, Fanglu Luo, Yichen Wang, Jun Cui
[Paper # NCOMMS-22-38439A]

Response to Reviewers

The reviewers' comments are in black Times New Roman font. My responses are in blue Arial.

REVIEWER COMMENTS

Reviewer #1 (Remarks to the Author):

The reviewer largely appreciates the authors. The reviewer confirmed that the authors adequately addressed all my concerns. The reviewer believes that the revised manuscript clearly demonstrates the significance of the non-trackable distal ejecta on the understanding the regional and global stratigraphy on planetary bodies. The reviewer notes only some very minor points for the final revision before submission (which I do not need to see again).

The authors appreciate the comments and suggestions kindly provided by this reviewer.

L253–254: “Spall fragments excavated by oblique impacts exhibit azimuth asymmetry in the ejection velocities and angles³⁵.” The reference #33 (oblique impacts) should be cited in this context, not #35 (vertical impacts only).

Changed.

L263–264: Sabuwala et al., 2018(#37) also conducted a series of laboratory impact experiments. Please cite the previous study more accurately.

Revised accordingly.

L270: What is “ample condition”? Is this typo?

“ample” is changed to “abundant”.

Figure 9b: The authors should indicate the locations of the faint impact rays with several arrows or something like that.

Arrows that indicate the locations of the faint impact rays are added.

Supplementary Figs. 6 and 7: The authors need to explain the meaning of the color in the figure. I recommend that a color bar would be added beside the figure.

Supplementary Fig. 6c shares the same color code with Supplementary Fig. 6b; Supplementary Fig. 7c shares the same color code with Supplementary

Fig. 7b. Color used in Supplementary Fig. 7a was referred from Lagain et al. (2021), which denotes spatial densities of impact craters at different diameter ranges. Such information is now elaborated in the figure legends.

Reviewer #2 Dr. Anthony Lagain (Remarks to the Author):

The authors have addressed each one of my concerns in detail. This revised manuscript is better presented, clearer, the implications well-justified, and study cases convincing. In my opinion the manuscript is ready for publication.

The authors appreciate the positive evaluation by Dr. Lagain.

Thank you again for all the comments and suggestions.

Responses completed.